# Weighted model estimation for offline model-based reinforcement learning

**Toru Hishinuma**
Kyoto University
`hishinuma.toru.43n@kyoto-u.jp`

**Kei Senda**
Kyoto University
`senda@kuaero.kyoto-u.ac.jp`

## Abstract

This paper discusses model estimation in offline model-based reinforcement learning (MBRL), which is important for subsequent policy improvement using an estimated model. From the viewpoint of covariate shift, a natural idea is model estimation weighted by the ratio of the state-action distributions of offline data and real future data. However, estimating such a natural weight is one of the main challenges for off-policy evaluation, which is not easy to use. As an artificial alternative, this paper considers weighting with the state-action distribution ratio of offline data and simulated future data, which can be estimated relatively easily by standard density ratio estimation techniques for supervised learning. Based on the artificial weight, this paper defines a loss function for offline MBRL and presents an algorithm to optimize it. Weighting with the artificial weight is justified as evaluating an upper bound of the policy evaluation error. Numerical experiments demonstrate the effectiveness of weighting with the artificial weight.

## 1 Introduction

Reinforcement learning (RL) is a framework to learn a policy in an unknown environment [1]. Model-based RL (MBRL) is an approach that explicitly estimates a transition model and utilizes it to improve a policy. Compared to model-free RL, one major advantage of MBRL is the data efficiency [2–4], which is important for applications where the data collection is expensive, such as robotics or healthcare. This paper focuses on offline MBRL, i.e. MBRL that learns a policy from previously collected datasets.

One of the main challenges in offline MBRL is distribution shift [5]. The distribution of offline data is different from the distribution of data obtained in the future when applying an improved policy. This is a situation called covariate shift, where a model estimated using standard supervised learning techniques such as empirical risk minimization (ERM) may not generalize to test data. A major idea in MBRL is to estimate a model using such a standard method and to improve a policy using the estimated model in regions where the predictions are accurate [6–8]. A point to be improved here is model estimation without considering covariate shift. In supervised learning under covariate shift, importance-weighted ERM can obtain good predictive performance [9]. If a model in offline MBRL can be estimated in a similar way, then the estimated model will make more accurate predictions, resulting in increased policy improvement. Motivated by this, this paper tackles the issue of weighted model estimation considering covariate shift in offline MBRL.

From the viewpoint of covariate shift, a natural idea is weighting with the ratio of the state-action distributions of offline data and real future data, as the agent uses a model to predict real future data. For convenience, this paper calls the ratio "the natural weight." However, estimating the natural weight is one of the main challenges for off-policy evaluation [5, 10]. Compared to density ratio estimation for supervised learning [11], the difficulty of off-policy evaluation is the inaccessibility of samples from the distribution of real future data.

35th Conference on Neural Information Processing Systems (NeurIPS 2021).

As an alternative, this paper considers weighting with the state-action distribution ratio of offline data and "simulated" future data. This paper calls the ratio "the artificial weight," in contrast to the natural weight. Simulated future data can be generated in offline simulation, unlike real future data. Since samples from both distributions are available, the artificial weight can be relatively easily obtained by standard density ratio estimation techniques for supervised learning. Based on the artificial weight, this paper defines a loss function for offline MBRL in Section 4. This paper presents an algorithm to optimize it in Section 5.

The question here is the validity of weighting with the artificial weight, as it may not be natural for covariate shift. This paper justifies it as evaluating an upper bound of the policy evaluation error in Section 4. This paper demonstrates that it works in practice in numerical experiments in Section 6. The above is the contribution of this paper.

## 2 Related Works

Most of offline MBRL researches [7, 8, 12] discuss how to learn a policy using a learned model. In contrast, this paper studies how to estimate a model considering covariate shift in MBRL, which depends on a policy to be applied. In principle, these can be combined as an algorithm that repeats between estimating a model and learning a policy, like Algorithm 3.

Importance-sampling-based methods estimate the natural weight, namely, the distribution ratio of offline data and real future data on a trajectory or state(-action) space [13–15, 10]. The advantage is unbiasedness, while the disadvantage is high variance. Model-based methods, including this paper, evaluate policies utilizing models trained using data. Model-based methods have the opposite advantage and disadvantage of importance-sampling-based methods. Doubly robust methods combine importance-sampling-based and model-based methods [16–18]. In principle, incorporating the idea of this paper as a model-based component is possible.

VAML [19, 20] defines a state-action-wise loss function based on the expected error of Bellman operator and learns a model to optimize a weighted sum. Similar ideas are found in [21–23]. The main focus is the state-action-wise loss function, while there are few studies about how to weight it is. The few studies, e.g. GAMPS [24], use the natural weight. In contrast, this paper simply uses the negative log likelihood loss function for a state-action pair and discusses the artificial weight, which is relatively easy to estimate.

This paper is similar to VMBPO [25] in that an EM-style algorithm updates a model and a policy using a universal loss function. The differences is that this paper considers covariate shift but does not use the idea of RL as inference [26]. Formulation that explicitly incorporates RL as inference is also an interesting topic.

Another approach is learning a model robust against covariate shift by reducing the difference between the distributions of offline data and future data on a feature space [22, 27, 28]. Since covariate shift measured in a feature space is encouraged to decrease, policy improvement increasing covariate shift is also discouraged. That is, such an idea implicitly limits policy improvement. This paper differs in that it is trying to utilize covariate shift.

## 3 Preliminaries

This paper discusses a discounted infinite horizon MDP [29]. Let $s$ be a state. Let $a$ be an action. Let $\rho(s)$ be an initial state distribution. Let $\mathcal{P}(s'|s, a)$ be a transition probability function. Let $r(s, a)$ be a reward function. Let $\gamma$ be a discount factor. Let $\pi$ be a policy. The value function is

$$V_{\mathcal{P}}^{\pi}(s^0) = \mathbb{E}_{a^t \sim \pi(\cdot|s^t), s_{t+1} \sim \mathcal{P}(\cdot|s^t, a^t)} \left[ \sum_{t=0}^{\infty} \gamma^t r(s^t, a^t) \middle| s^0 \right].$$

The expected return is $\eta_{\mathcal{P}}^{\pi} = \mathbb{E}_{s^0 \sim \rho} [V_{\mathcal{P}}^{\pi}(s)]$. The state distribution is

$$d_{\mathcal{P}}^{\pi}(s) = (1 - \gamma) \sum_{t=0}^{\infty} \gamma^t \Pr\left( s^t = s \middle| \rho, \pi, \mathcal{P} \right).$$

The state-action distribution is $d_{\mathcal{P}}^{\pi}(s, a) = d_{\mathcal{P}}^{\pi}(s)\pi(a|s)$. Note that $\eta_{\mathcal{P}}^{\pi} = \mathbb{E}_{(s,a) \sim d_{\mathcal{P}}^{\pi}}[\frac{r(s,a)}{1-\gamma}]$.

Let $\mathcal{P}_*$ denote the true transition probability, which is unknown to the agent. Let $\mathcal{P}_\theta$ denote the transition probability model parameterized by $\theta$. Let $\mathcal{D} = \{(s_n, a_n, s'_n)\}_{n=1}^N$ be offline data, which is obtained from $\mathcal{P}_*$. Let $d_{\mathcal{P}_*}^{\mathcal{D}}$ be the distribution of the input part of $\mathcal{D}$, i.e. $\{(s_n, a_n)\}_{n=1}^N$. Let $\hat{\mathcal{D}}_\theta^\pi$ be simulation data generated given $(\mathcal{P}_\theta, \pi)$ with state initialization with probability $1 - \gamma$, which is equal to sampling $(s, a)$ from $d_{\mathcal{P}_\theta}^\pi$. Let $c_\theta(s, a) = \mathbb{E}_{s' \sim \mathcal{P}_*(\cdot|s,a)} [-\ln \mathcal{P}_\theta(s'|s, a)]$ be the cross entropy between $\mathcal{P}_*(\cdot|s, a)$ and $\mathcal{P}_\theta(\cdot|s, a)$. Let $h(s, a) = \mathbb{E}_{s' \sim \mathcal{P}_*(\cdot|s,a)} [-\ln \mathcal{P}_*(s'|s, a)]$ be the self-entropy of $\mathcal{P}_*(\cdot|s, a)$.

# 4 Loss Function

This paper discusses the theory for the case where the reward function is known. For the unknown reward function, if the prediction of a reward model is bounded, the same loss functions can be derived, see Appendix.

## 4.1 Policy Evaluation

The goal of policy evaluation is estimating $\eta_{\mathcal{P}_*}^\pi$ when given $\pi$. For a model-based setting, the goal reduces to finding $\theta$ decreasing $|\eta_{\mathcal{P}_*}^\pi - \eta_{\mathcal{P}_\theta}^\pi|$, which is bounded as

$$
\begin{aligned}
|\eta_{\mathcal{P}_*}^\pi - \eta_{\mathcal{P}_\theta}^\pi| &\leq \frac{\gamma \mathbb{E}_{(s,a) \sim d_{\mathcal{P}_\theta}^\pi} \left[ \left| \sum_{s'} \left( \mathcal{P}_*(s'|s, a) - \mathcal{P}_\theta(s'|s, a) \right) V_{\mathcal{P}_*}^\pi(s') \right| \right]}{1 - \gamma} \\
&\leq \frac{B \mathbb{E}_{(s,a) \sim d_{\mathcal{P}_\theta}^\pi} [||\mathcal{P}_*(\cdot|s, a) - \mathcal{P}_\theta(\cdot|s, a)||_1]}{\sqrt{2}} \\
&\leq B \sqrt{\mathbb{E}_{(s,a) \sim d_{\mathcal{P}_\theta}^\pi} [c_\theta(s, a) - h(s, a)]} \\
&= B \sqrt{\mathbb{E}_{(s,a) \sim d_{\mathcal{P}_*}^\mathcal{D}} [w_\theta^\pi(s, a) c_\theta(s, a)] - \mathbb{E}_{(s,a) \sim d_{\mathcal{P}_*}^\mathcal{D}} [w_\theta^\pi(s, a) h(s, a)]} \\
&\leq B \sqrt{\mathbb{E}_{(s,a) \sim d_{\mathcal{P}_*}^\mathcal{D}} [w_\theta^\pi(s, a) c_\theta(s, a)] - h_{\min}}, \tag{1}
\end{aligned}
$$

where $B = \frac{\gamma \max_{s,a} |r(s,a)| \sqrt{2}}{(1-\gamma)^2}$, $w_\theta^\pi(s, a) = \frac{d_{\mathcal{P}_\theta}^\pi(s,a)}{d_{\mathcal{P}_*}^\mathcal{D}(s,a)}$, and $h_{\min} = \min_{s,a} h(s, a)$. The first inequality is the Telescoping lemma [21]. The second inequality holds from Holder's inequality and $||V_{\mathcal{P}}^\pi||_\infty \leq \frac{\max_{s,a} |r(s,a)|}{1-\gamma}$. The third inequality holds from Pinsker's inequality and Jensen's inequality. When trying to optimize the third inequality, $\mathbb{E}_{(s,a) \sim d_{\mathcal{P}_*}^\mathcal{D}} [w_\theta^\pi(s, a) h(s, a)]$ is difficult to handle, as $h(s, a)$ is unknown. The fourth inequality (1) excludes the difficult part from the optimization.

This paper considers decreasing the policy evaluation error by minimizing Equation (1). What is to be minimized is

$$
L^\pi(\theta) = \mathbb{E}_{(s,a) \sim d_{\mathcal{P}_*}^\mathcal{D}} [w_\theta^\pi(s, a) c_\theta(s, a)] = \mathbb{E}_{(s,a) \sim d_{\mathcal{P}_*}^\mathcal{D}, s' \sim \mathcal{P}_*(\cdot|s,a)} [-w_\theta^\pi(s, a) \ln \mathcal{P}_\theta(s'|s, a)].
$$

The empirical average of $L^\pi(\theta)$ is

$$
\hat{L}^\pi(\theta) = -\frac{1}{N} \sum_{n=1}^N w_\theta^\pi(s_n, a_n) \ln \mathcal{P}_\theta(s'_n|s_n, a_n), \tag{2}
$$

Equation (2) is a loss function weighted with $w_\theta^\pi(s, a)$, which is the artificial weight, namely, the distribution ratio of offline data $\mathcal{D}$ and simulated future data $\hat{\mathcal{D}}_\theta^\pi$. Thus, weighting with the artificial weight is justified as evaluating an upper bound of the policy evaluation error.

## 4.2 Policy optimization

The goal of policy optimization is finding $\pi$ increasing $\eta_{\mathcal{P}_*}^\pi$. In offline MBRL, the agent utilizes $\eta_{\mathcal{P}_\theta}^\pi$ for the optimization. A lower bound of $\eta_{\mathcal{P}_*}^\pi$ is

$$
\eta_{\mathcal{P}_*}^\pi \geq \eta_{\mathcal{P}_\theta}^\pi - |\eta_{\mathcal{P}_*}^\pi - \eta_{\mathcal{P}_\theta}^\pi| \geq \eta_{\mathcal{P}_\theta}^\pi - B \sqrt{L^\pi(\theta) - h_{\min}}.
$$

The first term evaluates the expected return by model $\mathcal{P}_\theta$, and the second penalizes the policy evaluation error.

Inspired from increasing $\eta^\pi_{\mathcal{P}_*}$ by maximizing the lower bound, this paper defines a generalization loss function for offline MBRL by

$$J(\theta, \pi) = -\eta^\pi_{\mathcal{P}_\theta} + B'\sqrt{L^\pi(\theta) - h_{\min}} \tag{3}$$

where $B' \in [0, B]$ is a tunable coefficient to scale the penalty for the policy evaluation error.

The agent optimizes an estimate of Equation (3), where empirical averages of $\eta^\pi_{\mathcal{P}_\theta}$ and $L^\pi(\theta)$ are weighted with the artificial weight. Similarly to the discussion in Section 4.1, weighting with the artificial ratio is justified as evaluating the expected return with the penalty for the policy evaluation error.

**Remark.** If using the Telescoping lemma with respect to $d^\pi_{\mathcal{P}_*}$ instead of $d^\pi_{\mathcal{P}_\theta}$, the policy evaluation error is similarly bounded as

$$|\eta^\pi_{\mathcal{P}_*} - \eta^\pi_{\mathcal{P}_\theta}| \leq B\sqrt{\mathbb{E}_{(s,a)\sim d^{\mathcal{D}}_{\mathcal{P}_*}}\left[\frac{d^\pi_{\mathcal{P}_*}(s,a)}{d^{\mathcal{D}}_{\mathcal{P}_*}(s,a)}c_\theta(s,a)\right] - h_{\min}}, \tag{4}$$

where $\frac{d^\pi_{\mathcal{P}_*}(s,a)}{d^{\mathcal{D}}_{\mathcal{P}_*}(s,a)}$ is the natural weight. There are two advantages to consider Equation (1) rather than Equation (4). The first is that the artificial weight can be more easily ontained, as described in Section 1. The second is that Equation (1) leads to a standard MDP optimization problem, as described in Section 5.2, while Equation (4) results in a more complicated optimization problem related to both $d^\pi_{\mathcal{P}_\theta}$ and $d^\pi_{\mathcal{P}_*}$.

## 5  Algorithm

### 5.1  Policy Evaluation

#### 5.1.1  Full Version

This paper considers a gradient-based optimization algorithm. The gradient of Equation (2) is

$$\nabla_\theta \hat{L}^\pi(\theta) = -\frac{1}{N}\sum_{n=1}^N w^\pi_\theta(s_n, a_n)\left\{\nabla_\theta \ln \mathcal{P}_\theta(s'_n|s_n, a_n) + \ln \mathcal{P}_\theta(s'_n|s_n, a_n)z^\pi_\theta(s_n)\right\},$$

where $z^\pi_\theta(s) = \nabla_\theta \ln d^\pi_{\mathcal{P}_\theta}(s)$, which can be estimated using an extension of LSDG [30], as described later. The artificial weight, $w^\pi_\theta$, can be estimated using a density ratio estimation method given $(D, \hat{D}^\pi_{\mathcal{P}_\theta})$. The gradient of Equation (2) can be estimated using them.

In practice, the computational efforts to estimate $(w^\pi_\theta, z_\theta)$ are non-negligible. This paper presents a method that iterates between estimating $(w^\pi_\theta, z_\theta)$ and updating $\theta$ locally. To locally update from $\theta^{(i)}$, this paper uses a local approximate loss function defined by

$$\hat{L}^\pi(\theta; \theta^{(i)}) = -\frac{1}{N}\sum_{n=1}^N w^\pi_{\theta^{(i)}}(s_n, a_n)\left\{\ln \mathcal{P}_\theta(s'_n|s_n, a_n) + (\theta - \theta^{(i)})\cdot\left(\ln \mathcal{P}_{\theta^{(i)}}(s'_n|s_n, a_n)z^\pi_{\theta^{(i)}}(s_n)\right)\right\}. \tag{5}$$

The gradient of Equation (5) with respect to $\theta$ can be obtained immediately, as $(w^\pi_{\theta^{(i)}}, z^\pi_{\theta^{(i)}})$ is fixed. For $\theta = \theta^{(i)}$, Equation (2) and Equation (5) are equal, and their gradients are also equal. That is, for $\theta = \theta^{(i)}$, the first-order necessary conditions for optimality are the same. In theory, if assuming that $(w^\pi_{\theta^{(i)}}, z^\pi_{\theta^{(i)}})$ is accurately estimated, the above-mentioned iterative method converges.

Importance-weighted ERM is consistent but can be unstable in practice [9], even when the weight is exactly known. Importance-weighted ERM weighted with $w^\pi_{\theta^{(i)}}$ can be more unstable due to the estimation and approximation errors of density ratio estimation of $w^\pi_{\theta^{(i)}}$. To stabilize the estimation of $\theta$, this paper replaces $w^\pi_{\theta^{(i)}}(s_n, a_n)$ in Equation (5) with $[w^\pi_{\theta^{(i)}}(s_n, a_n)]^\alpha$, where $\alpha = 0$ corresponds to ERM, and $\alpha = 1$ corresponds to importance-weighted ERM weighted with $w^\pi_{\theta^{(i)}}(s_n, a_n)$. That is, $\alpha$ is a hyperparameter controlling the trade-off between consistency and stability.

**Algorithm 1** Weighted model estimation for policy evaluation (full version).

1: **Input:** $\mathcal{D}$, $\pi$, $\alpha$, and $\mathcal{P}_{\theta^{(0)}}$.
2: **for** $i = 0, 1, 2, \cdots$ **do**
3:     Generate $\hat{\mathcal{D}}_{\theta^{(i)}}^{\pi}$, using simulation given $(\mathcal{P}_{\theta^{(i)}}, \pi)$.
4:     Estimate $w_{\theta^{(i)}}^{\pi}$, using density ratio estimation given $(\mathcal{D}, \hat{\mathcal{D}}_{\theta^{(i)}}^{\pi})$.
5:     Estimate $z_{\theta^{(i)}}^{\pi}$, using extended-LSDG given $(\mathcal{P}_{\theta^{(i)}}, \pi)$.
6:     Obtain $\mathcal{P}_{\theta^{(i+1)}}$, maximizing Equation (5) given $(\mathcal{D}, w_{\theta^{(i)}}^{\pi}, z_{\theta^{(i)}}^{\pi}, \alpha)$.
7: **end for**

---

**Algorithm 2** Weighted model estimation for policy evaluation (simplified version).

1: **Input:** $\mathcal{D}$, $\pi$, $\alpha$, and $\mathcal{P}_{\theta^{(0)}}$.
2: **for** $i = 0, 1, 2, \cdots$ **do**
3:     Generate $\hat{\mathcal{D}}_{\theta^{(i)}}^{\pi}$, using simulation given $(\mathcal{P}_{\theta^{(i)}}, \pi)$.
4:     Estimate $w_{\theta^{(i)}}^{\pi}$, using density ratio estimation given $(\mathcal{D}, \hat{\mathcal{D}}_{\theta^{(i)}}^{\pi})$.
5:     Obtain $\mathcal{P}_{\theta^{(i+1)}}$, maximizing Equation (7) given $(\mathcal{D}, w_{\theta^{(i)}}^{\pi}, \alpha)$.
6: **end for**

---

**Extending LSDG.** LSDG [30] uses the gradient of $\ln d_{\mathcal{P}}^{\pi}(s)$ with respect to policy parameters in model-free RL. This paper extends this idea to the gradient with respect to model parameters in MBRL. In a discounted MDP, $d_{\mathcal{P}_\theta}^{\pi}$ satisfies

$$d_{\mathcal{P}_\theta}^{\pi}(s') = (1 - \gamma)\rho(s') + \gamma \sum_{s,a} d_{\mathcal{P}_\theta}^{\pi}(s)\pi(a|s)\mathcal{P}_\theta(s'|s,a).$$

The gradient with respect to the $j$-th parameter, $\theta_j$, is

$$d_{\mathcal{P}_\theta}^{\pi}(s')z_{\theta_j}^{\pi}(s') = \sum_{s,a} d_{\mathcal{P}_\theta}^{\pi}(s)\pi(a|s)\mathcal{P}_\theta(s'|s,a)\left\{\gamma\nabla_{\theta_i}\ln\mathcal{P}_\theta(s'|s,a) + \gamma z_{\theta_j}^{\pi}(s)\right\}, \qquad (6)$$

where $z_{\theta_j}^{\pi}(s) = \nabla_{\theta_j}\ln d_{\mathcal{P}_\theta}^{\pi}(s)$. Equation (6) implies that $z_{\theta_j}^{\pi}$ is the value function satisfying a "forward" Bellman equation, where the transition probability function is $\mathcal{P}_\theta$, and the reward function is $\gamma\nabla_{\theta_i}\ln\mathcal{P}_\theta(s'|s,a)$. To estimate $z_{\theta_j}^{\pi}$, this paper extends LSDG by replacing the gradient of $\ln\pi(a|s)$ with $\gamma\nabla_{\theta_j}\ln\mathcal{P}_\theta(s'|s,a)$. The extended-LSDG estimates $z_{\theta_j}^{\pi}$ using simulation data generated using $(\mathcal{P}_\theta, \pi)$. Since some off-policy evaluation methods estimate the discounted state distribution ratio as a solution to another forward Bellman equation [14, 15, 5], their extensions could also be used instead of LSDG.

Algorithm 1 summarizes how to optimize $\theta$ for policy evaluation, described above.

### 5.1.2 Simplified version

The computationally expensive part in Algorithm 1 is estimating $z_{\theta^{(i)}}^{\pi}$. This requires estimating the same number of the value functions satisfying Equation (6) as the number of model parameters. This is unrealistic for neural networks with many parameters. As a practical simplification, this paper also considers replacing Equation (5) with

$$\hat{L}^{\pi}(\theta; \theta^{(i)}) = -\frac{1}{N}\sum_{n=1}^{N} w_{\theta^{(i)}}^{\pi}(s_n, a_n)\ln\mathcal{P}_\theta(s_n'|s_n, a_n). \qquad (7)$$

Algorithm 2 is the resulting simplified algorithm. The price of the simplification is the loss of convergence in theory, as the first-order necessary conditions are no longer the same. For this reason, the number of iterations should be specified. In practice, Algorithm 2 also works well, see Section 6.

### 5.2 Policy Optimization

This paper considers an EM-style optimization algorithm, where the E-step fixes $\pi$ and optimizes $\theta$, and the M-step fixes $\theta$ and optimizes $\pi$. Instead of optimizing Equation (3) directly, this paper uses a

---

**Algorithm 3** Model-based offline policy optimization based on weighted model estimation.

---

1: **Input:** $D$, $\alpha$, $B'$, $\mathcal{P}_{\theta^{(0)}}$, and $\pi^{(0)}$.
2: **for** $i = 0, 1, 2, \cdots$ **do**
3:   **if** $i = 0$ **then**
4:     Estimate $\mathcal{P}_{\theta^{(i+1)}}$, using ERM given $\mathcal{D}$.
5:   **else**
6:     Compute $\hat{b}$, given $(B', \mathcal{D}, \mathcal{P}_{\theta^{(i)}}, \pi^{(i)})$.
7:     Obtain $\mathcal{P}_{\theta^{(i+1)}}$, using E-step given $(\mathcal{D}, \hat{b}, \pi^{(i)}, \alpha)$.
8:   **end if**
9:   Estimate $\hat{c}_{\theta^{(i+1)}}$, using ERM given $(\mathcal{D}, \mathcal{P}_{\theta^{(i+1)}})$.
10:  Compute $\hat{b}$, given $(B', \mathcal{D}, \mathcal{P}_{\theta^{(i+1)}}, \pi^{(i)})$.
11:  Update $\pi^{(i+1)}$, using M-step given $(\mathcal{P}_{\theta^{(i+1)}}, \hat{b}, \hat{c}_{\theta^{(i+1)}})$.
12: **end for**

---

surrogate function in the majorization minimization framework [31],

$$
\begin{aligned}
J_{\text{surr}}(\theta, \pi; \theta^{(i)}, \pi^{(j)}) &= -\eta_\theta^\pi + b(L^\pi(\theta) - h_{\min}) + C \\
&= -\mathbb{E}_{(s,a) \sim d_{\mathcal{P}_\theta}^\pi} \left[ \frac{r(s,a)}{1 - \gamma} - b(c_\theta(s,a) - h_{\min}) \right] + C,
\end{aligned} \tag{8}
$$

where $b = \dfrac{B'}{2\sqrt{L^{\pi^{(j)}}(\theta^{(i)}) - h_{\min}}}$ and $C = b(L^{\pi^{(j)}}(\theta^{(i)}) - h_{\min})$. The advantages of using Equation (8) are that the E-step reduces to a similar problem with Section 5.1 and that the M-step comes down to a MDP optimization problem, as described later.

Let $\hat{b} = \dfrac{B'}{2\sqrt{\hat{L}^{\pi^{(j)}}(\theta^{(i)}) - \hat{h}_{\min}}}$ be an estimate of $b$, where $\hat{h}_{\min} = \min_{(s,a,s') \in \mathcal{D}} \left[ -\ln \mathcal{P}_{\theta^{(i)}}(s'|s,a) \right]$. Note that $\hat{L}^{\pi^{(j)}}(\theta^{(i)})$ is computed using Equation (2) given $(\mathcal{D}, \mathcal{P}_{\theta^{(i)}}, w_{\theta^{(i)}}^{\pi^{(j)}})$, where $w_{\theta^{(i)}}^{\pi^{(j)}}$ can be estimated using density ratio estimation given $(\mathcal{D}, \hat{\mathcal{D}}_{\theta^{(i)}}^{\pi^{(j)}})$, and $\hat{\mathcal{D}}_{\theta^{(i)}}^{\pi^{(j)}}$ can be generated using simulation given $(\mathcal{P}_{\theta^{(i)}}, \pi^{(j)})$. That is, $\hat{b}$ can be computed when given $(\mathcal{D}, \mathcal{P}_{\theta^{(i)}}, \pi^{(j)})$.

**E-step.** To update $\theta$ while fixing $\pi = \pi^{(i)}$, the E-step estimates Equation (8) by

$$
-\frac{1}{N} \sum_{n=1}^{N} w_\theta^{\pi^{(i)}}(s_n, a_n) \left[ \frac{r(s_n, a_n)}{(1 - \gamma)} + \hat{b} \ln \mathcal{P}_\theta(s_n'|s_n, a_n) \right], \tag{9}
$$

where the constant terms are ignored. Equation (9) can be minimized using Algorithm 1 replacing $\ln \mathcal{P}_\theta(s_n'|s_n, a_n)$ with $\ln \mathcal{P}_\theta(s_n'|s_n, a_n) + \frac{r(s_n, a_n)}{\hat{b}(1 - \gamma)}$, or Algorithm 2.

**M-step.** To update $\pi$ while fixing $\theta = \theta^{(i+1)}$, the M-step estimates Equation (8) by

$$
-\mathbb{E}_{(s,a) \sim d_{\mathcal{P}_{\theta^{(i+1)}}}^\pi} \left[ \frac{r(s,a)}{1 - \gamma} - \hat{b}(\hat{c}_{\theta^{(i+1)}}(s,a) - \hat{h}_{\min}) \right], \tag{10}
$$

where $\hat{c}_{\theta^{(i+1)}}$ is a model of $c_{\theta^{(i+1)}}$, and $C$ is ignored. This paper trains $\hat{c}_{\theta^{(i+1)}}$ using input data $\{(s_n, a_n)\}_{n=1}^N$ and output data $\{-\ln \mathcal{P}_{\theta^{(i+1)}}(s_n'|s_n, a_n)\}_{n=1}^N$. Note that Equation (10) is the expected return in a MDP, where the transition probability is $\mathcal{P}_{\theta^{(i+1)}}$, and the reward function is

$$
r(s,a) - (1 - \gamma)\hat{b}(\hat{c}_{\theta^{(i+1)}}(s,a) - \hat{h}_{\min}).
$$

Here, $\hat{c}_{\theta^{(i+1)}}(s,a) - \hat{h}_{\min}$ can be seen as an estimate of the modeling error, and $(1 - \gamma)\hat{b}$ can be seen as the coefficient to scale the penalty for the modeling error. Since this MDP is known to the agent, Equation (10) can be optimized using any MDP optimization approaches, including RL algorithms.

Algorithm 3 summarizes the above-mentioned procedure.

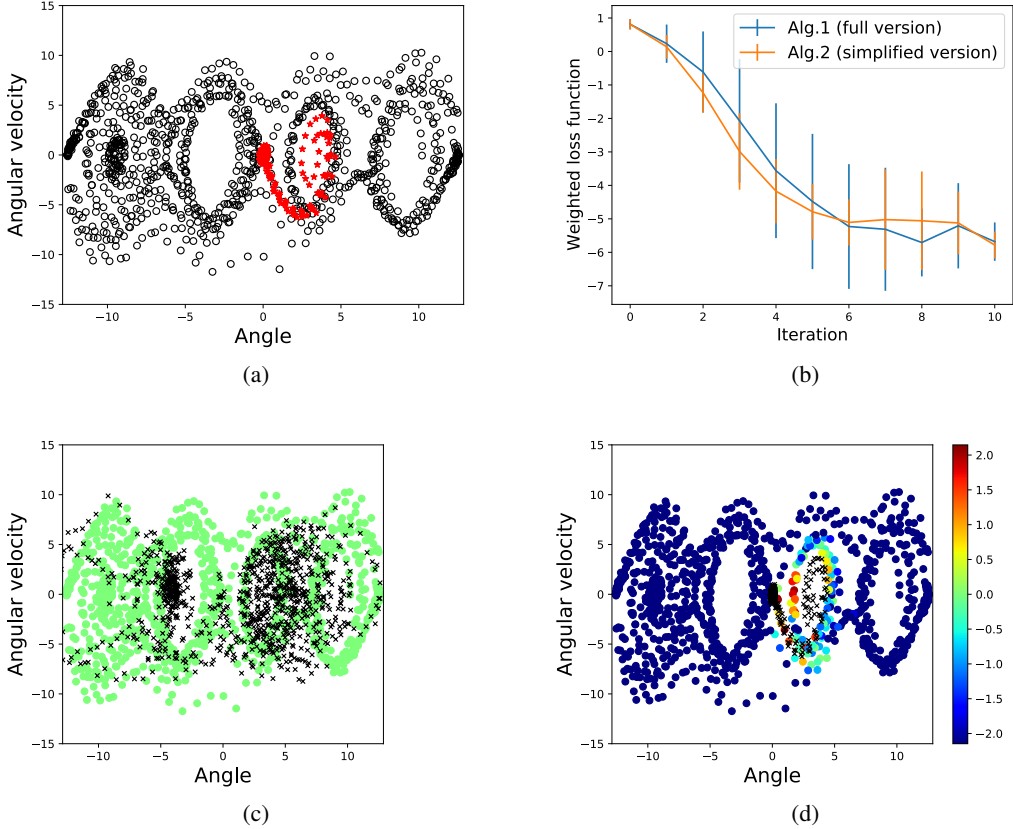

Figure 1: Pendulum task. (a) state of offline data and real future data, denoted by black circle and red star. (b) learning curves of weighted loss function. (c) simulated future states when using ERM. (d) simulated future states when using Algorithm 1. In (c) and (d), black crosses are simulated future states, and colormap shows $\log w_\theta^\pi(s_n, a_n)$ for offline data.

**Remark.** If $\alpha = 0$, then the E-step reduces to ERM, and Lines 6-7 in Algorithm 3 are skipped. This case can be seen as an instantiation of MOPO [8], where $\hat{c}_{\theta(i+1)} - \hat{h}_{\min}$ and $\hat{b}$ are the admissible error estimator and the penalty coefficient, respectively. This means that the M-step is essentially the same as MOPO, while the E-step is different from MOPO. In other words, MOPO can be seen applying only M-step to optimize the penalized return. Thus, Algorithm 3 improves MOPO in that it optimizes the penalized return with respect to both a model and a policy by the E-step and M-step.

## 6 Numerical Experiment

### 6.1 Pendulum

Firstly, to simply illustrate the effectiveness of weighting with the artificial weight, this paper studies policy evaluation on a pendulum environment. State $s$ is the pair of angle and angular velocity, and action $a$ is the torque. Offline data $\mathcal{D}$ is collected using a random policy. The initial state is around $(\pi, 0)$, and the cost-free state is $(0, 0)$. Target policy $\pi$ is a controller that swing-up the pendulum to $(0, 0)$ in the real environment. That is, real future data $\mathcal{D}_{\mathcal{P}_*}^\pi$ is the swing-up behavior towards $(0, 0)$. Figure 1 shows the state part of $\mathcal{D}$ and $\mathcal{D}_{\mathcal{P}_*}^\pi$.

The outline of the implementation is as follows. The agent uses $\mathcal{P}_\theta$ represented by two-layer neural networks with 8 units with tanh activation. The agent estimates $w_\theta^\pi$ using density ratio estimation via logistic regression [32, 11]. The agent estimates $z_\theta^\pi$ using an extension of LSDG [30]. For more details, see Appendix.

**Unweighted vs weighted ERM.** The discounted return of $\pi$ averaged over 100 episodes in the real environment is $-15.7$. This paper regards this value as the expected return, namely, the goal of policy evaluation. When using the ERM method, the estimated value is $-80.7$. When using Algorithm 1, the estimated value is $-17.8$. That is, Algorithm 1 estimates the expected return more accurately. Below, this paper describes why Algorithm 1 obtains such a better estimate.

Figure 1 (c) shows simulated future data $\hat{D}_{\mathcal{P}_\theta}^\pi$ when using the unweighted maximum likelihood method, a typical ERM method. The pendulum is predicted not to swing-up to $(0,0)$, unlike $D_{\mathcal{P}_*}^\pi$. This result can be described as follows. Based on the unweighted loss function, $\mathcal{P}_\theta$ tries to represent $\mathcal{D}$ as much as possible. However, its expressiveness is insufficient for global prediction. As a result, $\mathcal{P}_\theta$ does not predict the swing-up behavior.

Figure 1 (d) shows $\hat{D}_{\mathcal{P}_\theta}^\pi$ when using Algorithm 1 with $\alpha = 1$. The state part of $\mathcal{D}$ is colored based on the logarithm of $w_\theta^\pi(s_n, a_n)$ when Algorithm 1 converges. The pendulum is predicted to swing-up to $(0,0)$, like $D_{\mathcal{P}_*}^\pi$. This situation can be described as follows. When Algorithm 1 converges, Line 3 in Algorithm 1 predicts the swing-up behavior shown in Figure 1. Using the swing-up behavior as $\hat{D}_{\mathcal{P}_\theta}^\pi$, Line 4 estimates $w_\theta^\pi(s_n, a_n)$ shown in Figure 1. After estimating $z_\theta^\pi$ at Line 5, Line 6 estimates $\mathcal{P}_\theta$ to predict more accurately where the weight is larger. Since the expressive capability of $\mathcal{P}_\theta$ is collected near the larger weights, $\mathcal{P}_\theta$ can predict the same swing-up behavior again at Line 4 in the next iteration, despite the lack of global expressiveness as mentioned in the paragraph above. This is what happens when Algorithm 1 converges.

**Simplified version.** Figure 1 (b) shows the lerning curves of Algorithm 1 and Algorithm 2, where $\theta$ is initialized by the learning result of the ERM method, and $w_\theta^\pi(s_n, a_n)$ is normalized to satisfy $\frac{1}{N}\sum_n w_\theta^\pi(s_n, a_n) = 1$. The results are averaged over 5 runs, and the errorbar is the standard deviation. The simplified version also reduces the weighted loss function, compared to the ERM method, i.e. the start of the iteration. This result suggests that, although the simplified version has the limitation of not converging, it can be used as better than ERM. The simplified version can also predict the swing-up behavior, similarly to the full version, see Appendix.

## 6.2 D4RL MuJoCo Benchmark

Secondly, to demonstrate weighting with the artificial weight on large-scale environments, this paper studies policy optimization on the D4RL Benchmark [33] based on the MuJoCo simulator [34]. This paper shows the results of Algorithm 3 with $\alpha = 0$ and $\alpha = 0.2$. This paper regards the case of $\alpha = 0$ as a MBRL baseline, as it can be seen as an instantiation of MOPO [8]. Appendix discusses the result when using different values for $\alpha$ for the Walker2d-medium-expert dataset.

The outline of the implementation is as follows. The agent iterates the main loop of Algorithm 3 twice. The agent uses the simplified version of the E-step. The agent estimates $\Pr(s', r|s, a)$ instead of $\mathcal{P}(s'|s, a)$, to handle the unknown reward function, as in [8]. The agent represents $\Pr(s', r|s, a)$ using an ensemble of models, where each model is parametrized as 7-layer neural networks with 256 hidden units. The agent estimates $w_\theta^\pi$ in the same way as in Section 6.1. The agent estimates $c_\theta$ using L2 norm minimization. For the M-step, the agent improves $\pi$ using SAC [35], where samples are drawn from $\mathcal{D} \bigcup \hat{\mathcal{D}}_\theta^\pi$. This paper implements SAC by modifying the implementation code by [36]. To stabilize the M-step, the agent terminates a simulation episode when a state or action variable diverges. In addition, the agent clips a penalized reward so that the absolute value is not more than 10 times the maximum absolute value of the reward data. For more details, see Appendix.

Table 1 shows the normalized scores of Algorithm 3, where the results are averaged over 5 runs, and $\pm$ means the standard deviations. The major differences are found for Walker2d-medium-expert and Walker2d-medium datasets. The normalized scores of $\alpha = 0.2$ is better for the former and worse for the latter. Below, this paper describes the reasons.

**Walker2d-medium-expert dataset.** The case of $\alpha = 0.2$ moves forward without falling for all runs, while the case of $\alpha = 0$ falls down for two runs. Figure 2 (a) shows the learning curves of real and simulated returns in the M-step at the second iteration for the seed for that run. The case of $\alpha = 0.2$ obtains similar curves for real and simulation, while the case of $\alpha = 0$ does not. This difference comes from the weighted model estimation. In the case of $\alpha = 0$, the model for the first move is not sufficiently accurate, and this lead to falling at the first move. In the case of $\alpha = 0.2$,

Table 1: D4RL MuJoCo Benchmark: normalized scores. The scores of CQL and original MOPO are taken from [37] and [8].

| dataset | CQL [37] | original MOPO [8] | $\alpha = 0$ | $\alpha = 0.2$ |
|---|---|---|---|---|
| HalfCheetah-random | 35.4 | $35.4 \pm 2.5$ | $48.7 \pm 2.8$ | $49.1 \pm 3.2$ |
| HalfCheetah-medium | 44.4 | $42.3 \pm 1.6$ | $75.7 \pm 1.5$ | $73.1 \pm 5.2$ |
| HalfCheetah-medium-replay | 46.2 | $53.1 \pm 2.0$ | $72.1 \pm 1.4$ | $65.5 \pm 6.4$ |
| HalfCheetah-medium-expert | 62.4 | $63.3 \pm 38.0$ | $73.9 \pm 24.2$ | $85.7 \pm 21.6$ |
| Hopper-random | 10.8 | $11.7 \pm 0.4$ | $30.2 \pm 4.4$ | $32.7 \pm 0.5$ |
| Hopper-medium | 86.6 | $28.0 \pm 12.4$ | $100.9 \pm 2.7$ | $104.1 \pm 1.2$ |
| Hopper-medium-replay | 48.6 | $67.5 \pm 24.7$ | $97.2 \pm 10.9$ | $104.0 \pm 3.2$ |
| Hopper-medium-expert | 111.0 | $23.7 \pm 6.0$ | $109.3 \pm 1.1$ | $104.9 \pm 10.1$ |
| Walker2d-random | 7.0 | $13.6 \pm 2.6$ | $16.5 \pm 6.6$ | $18.4 \pm 7.6$ |
| Walker2d-medium | 74.5 | $17.8 \pm 19.3$ | $81.7 \pm 1.2$ | $60.7 \pm 29.0$ |
| Walker2d-medium-replay | 32.6 | $39.0 \pm 9.6$ | $80.7 \pm 3.1$ | $82.7 \pm 3.3$ |
| Walker2d-medium-expert | 98.7 | $44.6 \pm 12.9$ | $59.5 \pm 49.4$ | $108.2 \pm 0.5$ |

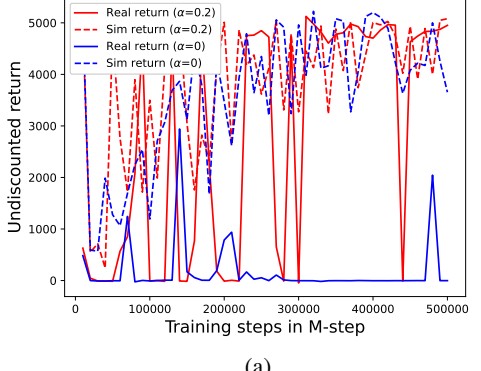 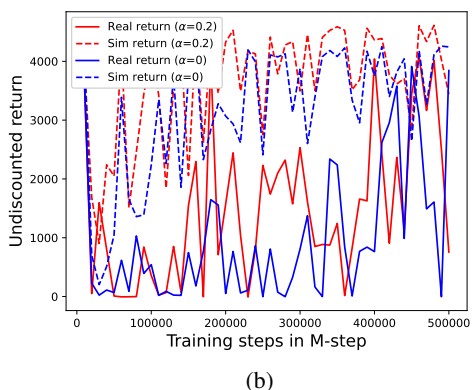

(a)               (b)

Figure 2: D4RL MuJoCo Benchmark: learning curves of (a) walker2d-medium-expert and (b) walker2d-medium.

simulated future data includes predictions of the first move, and similar parts of offline data are weighted with large weights. Based on the weights, the estimated model more accurately predicts the first move. As a result, the case of $\alpha = 0.2$ can learn to move forward without falling, based on the more accurate prediction. That is, Algorithm 3 improves the performance based on weighted model estimation. For the other runs, although the case of $\alpha = 0$ moves forward without falling in a real environment, it obtains real and simulation returns that are relatively different, see Appendix.

**Walker2d-medium dataset.** The case of $\alpha = 0.2$ falls down for two runs, while the case of $\alpha = 0$ moves forward without falling. Figure 2 (b) shows the learning curves fo those runs. In Figure 2 (b), for both $\alpha = 0$ and $\alpha = 0.2$, the simulation curves do not seem to capture the real returns. The difference of the score seems due to the high variance of the real returns.

# 7 Limitations

This paper has limitations common to standard offline MBRL approaches, such as Markov assumption and reward design. The limitations specific to this paper are as follows. This paper derives Algorithm 1 based on the theory, but it cannot be applied to large-scale problems due to the large amount of computations. This paper also presents Algorithm 2 as a practical simplified version, but it has no convergence with respect to Equation (3), see Section 5.1.

# 8 Conclusion and Future Direction

This paper discusses weighted model estimation considering covariate shift in offline MBRL. The main idea of this paper is weighting with the artificial weight, which is relatively easy to estimate. Based on this idea, this paper defines a loss function for offline MBRL and presents algorithms to optimize it. This paper justifies this idea as evaluating the upper bound of the policy evaluation error. This paper demonstrates the effectiveness of this idea in numerical experiments. The main contribution of this paper is showing the validity of weighting with the artificial weight.

The propose algorithm shows clear improvement in Section 6.1, while it improves only one case in Section 6.2. One reason for this gap might be that Section 6.1 needs interpolation, while Section 6.2 requires extrapolation. Although interpolation and near extrapolation can be addressed by importance-weighting, far extrapolation would require additional idea, e.g. using a structured prior. Since further discussion is needed, we leave it a future work.

Another interesting direction is extending to Bayesian MBRL [38, 39]. In supervised learning under covariate shift, the posterior distribution can be defined using a weighted likelihood function [9]. In offline MBRL, the likelihood function weighted with the artificial weight is effective, as shown in this paper. Combining these ideas, a weighted posterior considering covariate shift will be introduced into Bayesian MBRL.

## Acknowledgments and Disclosure of Funding

This work was partly supported by Collaborative Research for Evolutionary Mechanical System Technology with Mitsubishi Electric Corporation and by General Research Grant from SECOM Science and Technology Foundation. We thank all anonymous reviewers for their constructive comments.

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
