# Supplementary material

## 9   Extending to unknown reward function

Let $r_*$ be the real reward function. Let $r_\theta$ be a reward function model. Let $\eta^\pi(\mathcal{P}, r)$ be the discounted expected return of $\pi$ in a MDP whose transition and reward functions are $\mathcal{P}$ and $r$.

Using the triangle inequality, the policy evaluation error is bounded as

$$|\eta^\pi(\mathcal{P}_*, r_*) - \eta^\pi(\mathcal{P}_\theta, r_\theta)| \quad \leq \quad |\eta^\pi(\mathcal{P}_*, r_*) - \eta^\pi(\mathcal{P}_\theta, r_*)| + |\eta^\pi(\mathcal{P}_\theta, r_*) - \eta^\pi(\mathcal{P}_\theta, r_\theta)|$$

The first term is bounded as

$$
\begin{aligned}
|\eta^\pi(\mathcal{P}_*, r_*) - \eta^\pi(\mathcal{P}_\theta, r_*)| \quad &\leq \quad \frac{\gamma \mathbb{E}_{(s,a)\sim d_{\mathcal{P}_\theta}^\pi} \left[ \left| \sum_{s'} \left( \mathcal{P}_*(s'|s,a) - \mathcal{P}_\theta(s'|s,a) \right) V_{\mathcal{P}_*, r_*}^\pi(s') \right| \right]}{1-\gamma} \\[2mm]
&\leq \quad \frac{B \mathbb{E}_{(s,a)\sim d_{\mathcal{P}_\theta}^\pi} \left[ ||\mathcal{P}_*(\cdot|s,a) - \mathcal{P}_\theta(\cdot|s,a)||_1 \right]}{\sqrt{2}}
\end{aligned}
$$

where $V_{\mathcal{P}_*, r_*}^\pi$ is the value function in the real MDP. This derivation is the same with the main paper. That is, the first equality is the Telescoping lemma [1], and the second inequality holds from Holder's inequality and $||V_{\mathcal{P}_*, r_*}^\pi||_\infty \leq \frac{\max_{s,a} |r(s,a)|}{1-\gamma}$. Thus, the first term is bounded by Equation (4).

The second term is bounded as

$$
\begin{aligned}
|\eta^\pi(\mathcal{P}_\theta, r_*) - \eta^\pi(\mathcal{P}_\theta, r_\theta)| \quad &= \quad |\mathbb{E}_{(s,a)\sim d_{\mathcal{P}_\theta}^\pi}[r_*(s,a) - r_\theta(s,a)]| \\[2mm]
&\leq \quad \frac{||d_{\mathcal{P}_\theta}^\pi(s,a)||_1 ||r_*(s,a) - r_\theta(s,a)||_\infty}{1-\gamma} = \frac{||r_*(s,a) - r_\theta(s,a)||_\infty}{1-\gamma} \\[2mm]
&\leq \quad \frac{||r_*(s,a)||_\infty + ||r_\theta(s,a)||_\infty}{1-\gamma} = \frac{\max_{s,a} |r_*(s,a)| + \max_{s,a} |r_\theta(s,a)|}{1-\gamma}
\end{aligned}
$$

where the first inequality is Holder's inequality, and the second is the triangle inequality. If the prediction of a reward model is bounded by some constant, i.e. if $\max_{s,a} |r_\theta(s,a)| < C$, then the second term is bounded by $\frac{C + \max_{s,a} |r_*(s,a)|}{1-\gamma}$.

Since the first and second terms are bounded by Equation (4) and some constant, respectively, Equation (4) can also be used for reducing the policy evaluation error when the reward function is unknown. The discussion for policy optimization is the same.

There may be room for improving the upper bound for the second term, while optimizing the improved upper bound should become more complicated. This paper leaves it a future issue.

# 10  Extending LSDG

LSDG($\lambda$) [2] discusses the derivatives of $d_{\mathcal{P}}^{\pi}$ with respect to policy parameters. This paper applies the idea to the derivatives with respect to model parameters.

## 10.1  Preliminary

Let $p_f(a, s'|s) = \pi(a|s)\mathcal{P}(s'|s, a)$ be the probability of the succeeding action and state. The state value function satisfies

$$V_{\mathcal{P}}^{\pi}(s) = \mathbb{E}_{(a,s')\sim p_f(\cdot|s)}\left[r(s, a, s') + \gamma V_{\mathcal{P}}^{\pi}(s')\right]$$

Let $\xi$ be the value function approximation parameter vector. Most temporal difference methods such as LSTD [3] minimize the mean squared projected Bellman error (MSPBE) [4]. When using linear function approximation $V_{\mathcal{P}}^{\pi}(s) = \xi^T \phi(s)$, MSPBE is written as

$$\text{MSPBE} = (A\xi - b)^T C^{-1} (A\xi - b)$$

where

$$
\begin{aligned}
A &= \mathbb{E}_{s\sim d_{\mathcal{P}}^{\pi},(a,s')\sim p_f(\cdot|s)}\left[\phi(s)(\phi(s) - \gamma\phi(s'))^T\right] \\
b &= \mathbb{E}_{s\sim d_{\mathcal{P}}^{\pi},(a,s')\sim p_f(\cdot|s)}\left[r(s, a, s')\phi(s)\right] \\
C &= \mathbb{E}_{s\sim d_{\mathcal{P}}^{\pi}}\left[\phi(s)\phi(s)^T\right].
\end{aligned}
$$

For the derivation, see Equation (43) in [5]. MSPBE is minimized by $\xi = A^{-1}b$, where $A$ and $b$ can be estimated using LSTD.

## 10.2  Extension

In a discounted MDP, $d_{\mathcal{P}_\theta}^{\pi}$ satisfies

$$d_{\mathcal{P}_\theta}^{\pi}(s') = (1 - \gamma)\rho(s') + \gamma \sum_{s,a} d_{\mathcal{P}_\theta}^{\pi}(s)\pi(a|s)\mathcal{P}_\theta(s'|s, a).$$

The derivative with respect to the $i$-th model parameter component $\theta_i$ is

$$
\begin{aligned}
\nabla_{\theta_i} d_{\mathcal{P}_\theta}^{\pi}(s') &= \nabla_{\theta_i}\left[\gamma \sum_{s,a} d_{\mathcal{P}_\theta}^{\pi}(s)\pi(a|s)\mathcal{P}_\theta(s'|s, a)\right] \\
d_{\mathcal{P}_\theta}^{\pi}(s')\nabla_{\theta_i}\ln d_{\mathcal{P}_\theta}^{\pi}(s') &= \gamma \sum_{s,a} d_{\mathcal{P}_\theta}^{\pi}(s)\pi(a|s)\mathcal{P}_\theta(s'|s, a)\left[\nabla_{\theta_i}\ln d_{\mathcal{P}_\theta}^{\pi}(s) + \nabla_{\theta_i}\ln\mathcal{P}_\theta(s'|s, a)\right] \tag{18}
\end{aligned}
$$

Using the Bayes rule, the backward probability of the preceding state and action is

$$\tilde{p}_b(s, a|s') = \frac{p(s'|s, a)p(s, a)}{p(s')} = \frac{\mathcal{P}_\theta(s'|s, a)d_{\mathcal{P}_\theta}^{\pi}(s)\pi(a|s)}{d_{\mathcal{P}_\theta}^{\pi}(s')}$$

Using the backward probability, Equation (18) is rewritten as

$$\nabla_{\theta_i}\ln d_{\mathcal{P}_\theta}^{\pi}(s') = \gamma \sum_{s,a} \tilde{p}_b(s, a|s')\left[\nabla_{\theta_i}\ln\mathcal{P}_\theta(s'|s, a) + \nabla_{\theta_i}\ln d_{\mathcal{P}_\theta}^{\pi}(s)\right]$$

For short, using $z_i(s) = \nabla_{\theta_i}\ln d_{\mathcal{P}_\theta}^{\pi}(s)$ and $r'_i(s, a, s') = \gamma\nabla_{\theta_i}\ln\mathcal{P}_\theta(s'|s, a)$, the above equation is rewritten as

$$z_i(s') = \sum_{s,a} \tilde{p}_b(s, a|s')\left[r'_i(s, a, s') + \gamma z_i(s)\right] = \mathbb{E}_{(a,s)\sim\tilde{p}_b(\cdot|s')}\left[r'_i(s, a, s') + \gamma z_i(s)\right]$$

This is the value function under the backward Markov chain and the reward function $r'_i$. Note that, in the backward Markov chain, the state transition is from $s'$ to $s$.

When using linear function approximation $z_i(s) = \xi^T\phi(s)$, MSPBE for learning $z_i$ is

$$\text{MSPBE'} = (A'\xi - b')^T C'^{-1}(A'\xi - b')$$

where

$$A' = \mathbb{E}_{s' \sim \tilde{d}_{\mathcal{P}}^{\pi}, (s,a) \sim \tilde{p}_b(\cdot|s')} \left[ \phi(s')(\phi(s') - \gamma\phi(s))^T \right]$$

$$b' = \mathbb{E}_{s' \sim \tilde{d}_{\mathcal{P}}^{\pi}, (s,a) \sim \tilde{p}_b(\cdot|s')} \left[ r'_i(s,a,s')\phi(s') \right]$$

$$C' = \mathbb{E}_{s' \sim \tilde{d}_{\mathcal{P}}^{\pi}} \left[ \phi(s')\phi(s')^T \right]$$

and $\tilde{d}_{\mathcal{P}_\theta}^{\pi}(s)$ is the discounted state distribution under the backward Markov chain. Using $\tilde{d}_{\mathcal{P}}^{\pi}(s) = d_{\mathcal{P}}^{\pi}(s)$ (disconted MDP version of Proposition 1 in [2]), $A'$ can be rewritten as

$$
\begin{aligned}
A' &= \sum_{s,a,s'} \tilde{d}_{\mathcal{P}}^{\pi}(s')\tilde{p}_b(s,a|s')\phi(s')(\phi(s') - \gamma\phi(s))^T \\
&= \sum_{s,a,s'} d_{\mathcal{P}}^{\pi}(s')\tilde{p}_b(s,a|s')\phi(s')(\phi(s') - \gamma\phi(s))^T \\
&= \sum_{s,a,s'} d_{\mathcal{P}}^{\pi}(s)p_f(a,s'|s)\phi(s')(\phi(s') - \gamma\phi(s))^T \\
&= \mathbb{E}_{s \sim d_{\mathcal{P}}^{\pi}, (a,s') \sim p_f(\cdot|s)} \left[ \phi(s')(\phi(s') - \gamma\phi(s))^T \right]
\end{aligned}
$$

Similarly, $b'$ and $C'$ can be rewritten as

$$
\begin{aligned}
b' &= \mathbb{E}_{s \sim d_{\mathcal{P}}^{\pi}, (a,s') \sim p_f(\cdot|s)} \left[ r_i(s',a,s)\phi(s') \right] \\
C' &= \mathbb{E}_{s' \sim d_{\mathcal{P}}^{\pi}} \left[ \phi(s')\phi(s')^T \right] = C
\end{aligned}
$$

Note that $(A, b)$ and $(A', b')$ have the same form, except that $s$ and $s'$ are the opposite. This means that $(A', b')$ can be estimated using LSTD($\lambda$), replacing $s$ with $s'$.

The following equation is derived from $\sum_s d_{\mathcal{P}_\theta}^{\pi}(s) = 1$,

$$0 = \nabla_{\theta_i} \sum_s d_{\mathcal{P}_\theta}^{\pi}(s) = \sum_s d_{\mathcal{P}_\theta}^{\pi}(s)\nabla_{\theta_i} \ln d_{\mathcal{P}_\theta}^{\pi}(s) = \sum_s d_{\mathcal{P}_\theta}^{\pi}(s)z_i(s) = \mathbb{E}_{s \sim sd_{\mathcal{P}_\theta}^{\pi}}[\xi^T \phi(s)].$$

This can be seen as a constraint [2]. The violation of the constraint can be measured by $\xi^T D\xi$, where $D = \mathbb{E}_{s \sim sd_{\mathcal{P}_\theta}^{\pi}}[\phi(s)]\mathbb{E}_{s \sim sd_{\mathcal{P}_\theta}^{\pi}}[\phi(s)]^T$.

Finally, adding the violation penalty to MSPBE for leaning $z_i$, the loss function is

$$(A'\xi - b')^T C^{-1}(A'\xi - b') + \chi\xi^T D\xi$$

where $\chi$ is the coefficient for the violation penalty. This can be minimized by

$$\xi = (A'^T C^{-1}A' + \xi D)^{-1}A'^T C^{-1}b'.$$

The extension of LSDG($\lambda$) estimates this equation.

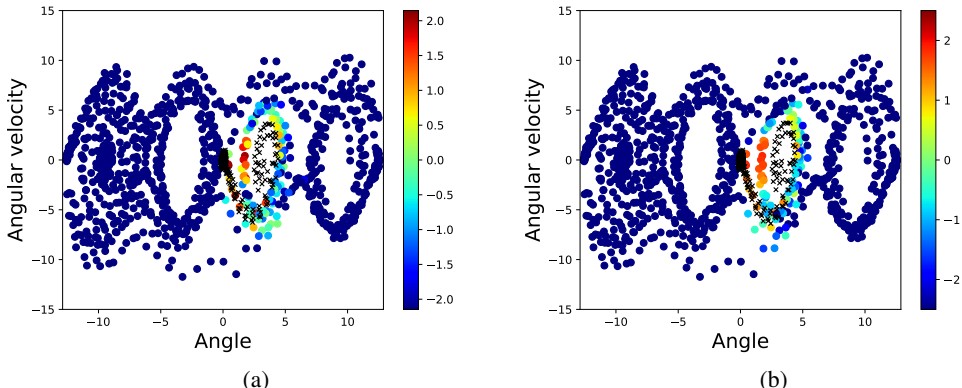

Figure 1: Pendulum task. (a) simulated future states when using Algorithms 1 (the same figure in the main paper), and (b) simulated future states when using Algorithms 2, where black crosses are simulated future states, and colormap show $\log w_\theta^\pi(s_n, a_n)$ for offline data.

## 11 Pendulum

### 11.1 Setting

Let $x$ and $v$ be the angle and angular velocity. The state is $s = (x, v)$. This task considers the bounded state space, $x \in [-4\pi, 4\pi]$ and $v \in [-15, 15]$. Let $u \in [-2.5, 2.5]$ be the underactuated torque. The transition function is the equation of motion of the pendulum,

$$v' = v + \frac{3}{m\ell^2}\left(u + mg\ell \sin(x) - cv\right)\Delta t$$
$$x' = x + v'\Delta t$$

where $m = 1$, $\ell = 1$, $g = 10$, $c = 0.01$, and $\Delta t = 0.1$. The reward function is

$$r(s, a) = -(1. - \exp(-x^2))$$

The target policy is trained using SAC [6] in the environment.

The agent represents $\mathcal{P}_\theta$ using two layer neural networks with 8 hidden units with tanh activation. The agent estimates $\mathcal{P}_\theta$ using the unweighted/weighted likelihood loss. The agent represents $c_\theta$ using two-layer neural networks with 16 units with tanh activation. The agent estimates $c_\theta$ using L2 loss. The agent represents $w_\theta^\pi$ using two-layer neural networks with 16 units with tanh activation. The agent estimates $w_\theta^\pi$ using the logistic regression loss [7]. The agent represents $z_\theta^\pi$ using linear function approximation, where the feature vector is $\phi(s) = (1, x, x^2, v, v^2)$. The agent estimates $z_\theta^\pi$ using the extension of LSDG($\lambda$) (see Section 10), where $\lambda = 0.9$ and $\chi = 0.1$.

### 11.2 Result of simplified version

Figure 1(b) shows $\hat{D}_{\mathcal{P}_\theta}^\pi$ when using Algorithm 2 with $\alpha = 1$, i.e. the simplified version. Similarly to Algorithm 1, the simplified version can also capture the stabilization behavior. When using Algorithm 2, the estimated value is $-17.7$, which is similar to when using Algorithm 1.

## 12 D4RL MuJoCo Benchmark

This paper uses two desktop PCs, with GeForce RTX 2060 SUPER and GeForce RTX 2070 SUPER.

### 12.1 Setting

This paper implements Algorithm 3 as follows. The main loop of Algorithm 3 is iterated twice.

**E-step.** The E-step is Algorithm 2, where $\theta$ is updated once per E-step. The agent estimates $\Pr(s', r|s, a)$ instead of $\mathcal{P}(s'|s, a)$, to handle the unknown reward function. The structure of $\mathcal{P}(s'|s, a)$ is the same as in the previous works [8–10], and only the model size and hyperparameter values are different. The agent represents $\mathcal{P}_\theta$ using a probabilistic ensemble of 7 models and pick the best 5 models, based on the log likelihod loss function weighted with $w_\theta^\pi$. Each of the model in the ensemble is parametrized as seven-layer neural networks with 256 hidden units and with swish activation. The learning rate is $10^{-3}$. The agent represents $w_\theta^\pi$ using five-layer neural networks with 256 hidden units and with tanh activation, based on the logistic regression loss [7]. The learning rate is $10^{-4}$. The agent represents $c_\theta^\pi$ using five-layer neural networks with 256 hidden units and with tanh activation, based on the log linear regression. The learning rate is $5 \times 10^{-3}$. Throughout the E-step, fitting is terminated by early-stopping with 10 percent of holdout data, and the batch size is 512.

**M-step.** The discounted factor is $\gamma = 0.995$. The penalty coefficient is $B' = 10^{-2}$. The agent improves $\pi$ using SAC [6], where samples are drawn from $\mathcal{D} \bigcup \hat{\mathcal{D}}_\theta^\pi$. The batch size is 512, where the numbers of samples of $\mathcal{D}$ and $\hat{\mathcal{D}}_\theta^\pi$ are 256 and 256. Policy $\pi$ is a gaussian policy represented using three-layer neural networks with 256 hidden units and with relu activation. The learning rate is $10^{-3}$. The entropy regularization coefficient is $0.2$. The number of times to perform SAC update per M-step is $5 \times 10^5$. This paper implements SAC by modifying the implementation code by [11]. To stabilize the M-step, the agent terminates a simulation episode when a state or action variable diverges. In addition, the agent clips a penalized reward so that the absolute value is not more than 10 times the maximum absolute value of the reward data.

**Performance measure.** In the M-step, the simulation and real undiscounted returns are evaluated averaging over 5 episodes per $10^4$ SAC updates, respectively. The figures of the lerning curves of the second iteration in the main paper show them. The normalized scores in Table 1 are computed using the last values of the learning curves of real returns, averaging over 5 runs. The $\pm$ in Table 1 means the standard deviation.

Table 1: Weighted loss function for policy evaluation error on Walker2d-medium-expert dataset (lower is better).

| run | $\alpha = 0.0$ | $\alpha = 0.1$ | $\alpha = 0.2$ | $\alpha = 0.5$ | $\alpha = 1.0$ |
|-----|------|------|------|------|------|
| (a) | -59.8 | -74.1 | -68.6 | -59.0 | -14.1 |
| (b) | -69.3 | -75.3 | -74.0 | -61.3 | -34.6 |
| (c) | -59.6 | -52.0 | -80.9 | -52.2 | -23.2 |
| (d) | -64.3 | -65.9 | -64.7 | -55.7 | 2112.4 |
| (e) | -68.0 | -67.8 | -66.6 | -53.8 | 202.5 |

## 12.2 Effect of $\alpha$

The newly introduced hyperparameter in this paper is $\alpha$, which interpolates between ERM and importance-weighted ERM, see Section 5. The purpose of $\alpha$ is to stabilize importance-weighted ERM in the E-step. The closer the $\alpha$ is to 0, the more stable the model estimation is, while the smaller the correction for covariate shift is. Below, this papers experimentally shows the effect of $\alpha$.

Figure 2 shows the learning curves for $\alpha = 0$, $\alpha = 0.1$, $\alpha = 0.2$, $\alpha = 0.5$, and $\alpha = 1$, on Walker2d-medium-expert dataset. The case of $\alpha = 1$ is poor in most cases. The case of $\alpha = 0$ obtains bad real returns in runs (b) and (d). The cases of $\alpha = 0.5$, $\alpha = 0.2$, and $\alpha = 0.1$ are relatively good in terms of real returns. This result shows that $\alpha$ controls the trade-off between consistency and stability.

Table 1 shows $\hat{L}^\pi(\theta)$ for learned policies in Figure 2, namely, the weighted loss functions for policy evaluation error. The case of $\alpha = 1$ is poor in all cases. The case of $\alpha = 0.5$ is also relatively bad. The case of $\alpha = 0.2$ is the best for run (c) in terms of $\hat{L}^\pi(\theta)$, and it is only the case where both real and simulated returns are good. The case of $\alpha = 0.1$ is the best for run (a), (b), and (d) in terms of $\hat{L}^\pi(\theta)$, and it has good real and simulated returns. The case of $\alpha = 0$ is the best for run (e) in terms of $\hat{L}^\pi(\theta)$, and it has good real and simulated returns in this run. Here, when $\hat{L}^\pi(\theta)$ is good, the results of real and simulated returns also tend to be good.

How to choose $\alpha$ is an important question, as the real retutn is unknown before applying a learned policy. For policy evaluation, Table 1 suggests that comparing $\hat{L}^\pi(\theta)$ gives an insight. For policy optimization, evaluating the estimate of $J(\pi, \theta)$ may be a more straightforward to policy optimization, while it also depends on another hyperparameter, $B'$, which scales the penalty of policy evaluation error. Note that $B'$ corresponds to the hyperparameter previously introduced in MOPO [10]. Since evaluating the estimate of $J(\pi, \theta)$ requires choosing $\alpha$ and $B'$ at the same time, it may be more complicated. Compared to it, this paper considers that comparing $\hat{L}^\pi(\theta)$ is a reasonable choice.

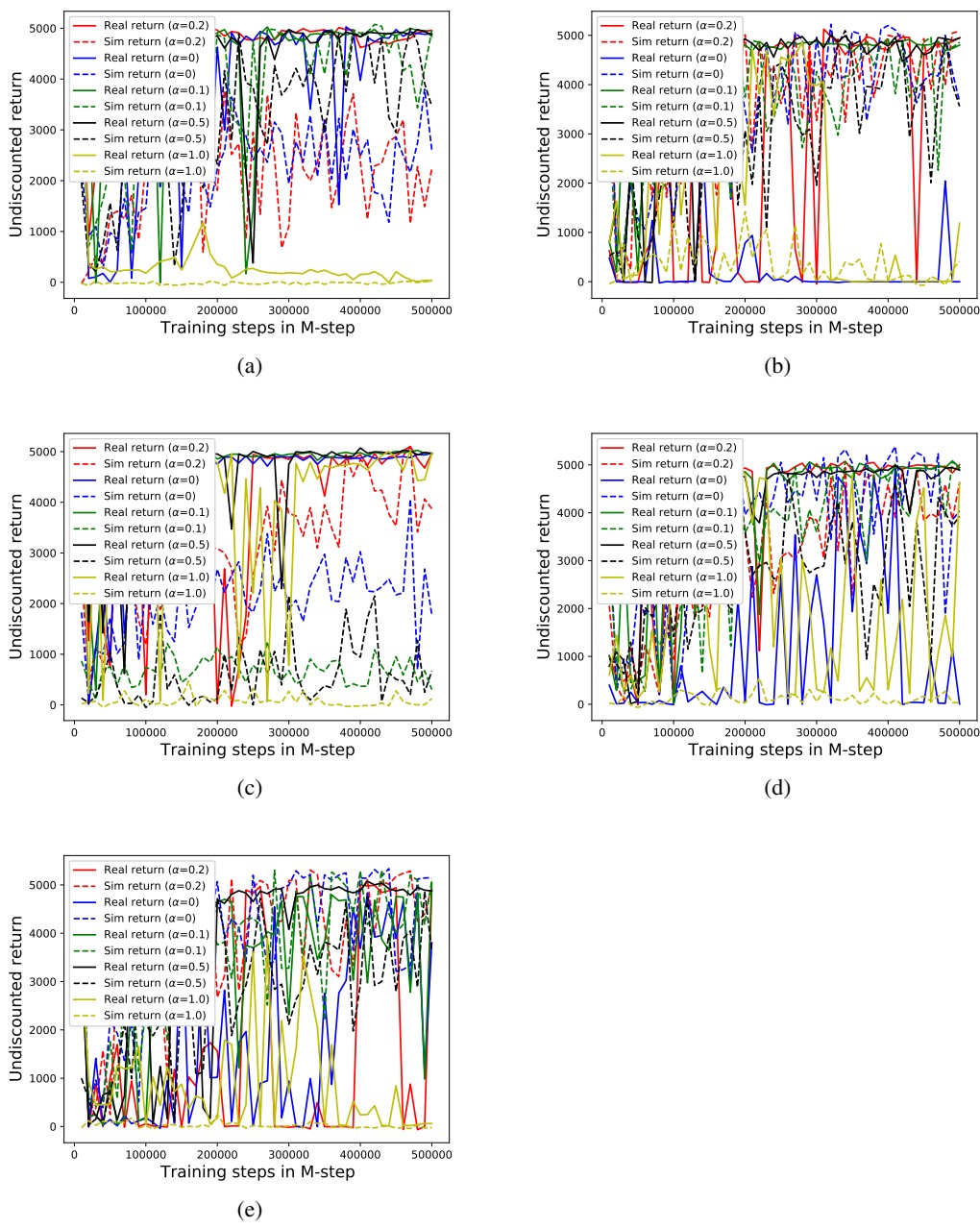

Figure 2: D4RL MuJoCo Benchmark. 5 runs on Walker2d-medium-expert dataset.