# OpenReview forum: "Weighted model estimation for offline model-based reinforcement learning"
_NeurIPS.cc/2021/Conference — NeurIPS 2021 Poster_

### Official Review · Reviewer_PdDh · 2021-07-05

**Rating:** 6
**Confidence:** 4

**Summary:**

The paper addresses the problem of covariate shift in offline memory-based reinforcement learning by proposing a novel method for more accurate evaluation of the expected return of a given policy by means of simulation of a learned model of the real system that might not be entirely correct. Two algorithm that uses a weighting factor, called the artificial weight, are proposed for tuning the learned system's model to predict more accurately the parts of the state space visited by the policy being evaluated, and thus improve the estimate of the expected return of this policy. This improved estimate can further be used for policy improvement, as customary for RL algorithms. An empirical evaluation in simulated environments demonstrates the validity of the proposed approach, both for policy evaluation and for policy improvement.

Update: I appreciate the clarification about the evaluation task the authors are solving (pendulum swing-up, not stabilization). I also noticed that the other reviewers also found the advantages of the proposed algorithm not all that overwhelming. This, combined with the generally unclear presentation, makes the paper borderline. I am revising my rating to 6, just above borderline - it would still be good to accept this paper, if possible, and depending on the ratings of other papers.

**Limitations And Societal Impact:**

I am not aware of any potential negative social impact of this work.

**Main Review:**

The method applies to the case when a data set is collected from a real system by means of an exploration policy, and the new policy to be evaluated would result in a significantly different state occupancy distribution than that of the exploration policy. In cases when the new policy is expensive to evaluate on the new system, and it is instead desired to be evaluated in simulation, using a model learned from the observed data that might have limited modeling capacity, this difference in the state distributions, combined with the possibly incorrect model of the dynamics, would cause incorrect prediction of the system trajectory under the policy, and from there, and incorrect estimate of its expected return on the real system.

As a solution to this problem, the paper proposes to use a weighting factor called artificial weight, expressed as a ratio of state-action occupancy probabilities, in order to iteratively tune the parameters encoding the dynamics of the learned system model to more accurately predict regions of state space where the policy being evaluated would bring the system to, and from there, the estimate of the expected return of the policy on the real system. In essence, the model of the dynamics ends up focusing on the parts of state space that will actually be used by the evaluated policy, instead of trying to model the dynamics everywhere. A parameter tuning algorithm with convergence guarantees is proposed, along with a simplified version that does not have such guarantees, but is computationally faster. For stability reasons, only a scaled-down value of the weight is used. Both versions of the algorithm are empirically evaluated on a pendulum stabilization problem and policy, and it is evident that the predicted trajectories and the estimate of the expected return of the policy are much closer to that of the real system that what a system model optimized by empirical risk minimization would produce.

Moreover, the improved estimate of the policy's return can further be used for policy improvement, as customary for RL algorithms. Empirical evaluation on the D4RL benchmark suite demonstrates improved performance with respect to the unweighted method. (Although, the comparison is not overwhelmingly in favor of the new method.)

Overall, the idea appears to be sound, and addresses a major issue with offline MBRL, which itself is a very important RL method for use cases that involve real physical equipment that is difficult, expensive, and slow to operate. It is highly original, and very promising with respect to solving challenging decision and control problems involving physical systems. The proposed work builds upon a lot of related research in ML concerning covariate shift and offline MBRL that is well discussed and referenced by the authors, and the combination of these ideas into the proposed algorithms is the main contribution of this paper. Based on this, I think the paper is a solid contribution to a field of high practical importance and significance, and thus merits acceptance.

At the same time, the presentation of the ideas and results could be improved. I found the paper fairly difficult to read. The iterative improvement of the parameter $\theta$ representing the system's dynamics is not explained very well. For example, in Algorithm 1, there is an iteration over index i, but it is not clear how many iterations there are, and what i actually indexes. Is it the successive estimates $\theta_i$ in lines 4, 5, and 6, respectively $\theta_{i+1}$ in line 7? If yes, Eq. 10 should probably be updated for consistency, too, replacing $\theta_0$ with $\theta_i$ and $\theta$ with $\theta_{i+1}$

It is also not entirely clear what the objective of the pendulum task is. Is it the pendulum stabilization task around the vertical position (and if yes, which one - up or down), or the much harder problem of pendulum swing-up? The goal state is angle 0 with 0 angular velocity, but does this angle correspond to the unstable up position, or the stable down position? I assume it is the former, but please clarify. It doesn't help that the starting state is not clearly shown, either. The authors say that they are solving the pendulum stabilization task, which is normally defined for relatively small angles around the unstable equilibrium, typically +/-15 degrees. However, Fig. 1 shows a trajectory in phase space (or is it two or more?) starting about pi radians away from the goal state (maybe in the stable equilibrium?) that pumps energy into the system, strongly suggesting a swing-up task with underactuated pendulum. Please clarify the task that the evaluated policy is trying to solve.

Minor typos:
Line 40: "the both" -> "both"
Line 74: "Preliminary" -> "Preliminaries"
Line 75: "a action" -> "an action"
Line 193: "with normalizing it" -> "and normalizing it"
Line 201: "common logarithm" -> "natural logarithm"?
Line 228: "the" -> "The"
Line 236: "averaged averaged" -> "averaged"
Line 237: "stanrard" -> "standard"



**Time Spent Reviewing:**

5

---

> ### Author Response · Authors · 2021-08-10
> **Response to Reviewer PdDh**
>
> Thank you for your helpful comment.
>
> **>>parameter $\theta$**
>
> The number of iterations is pre-specified, or selected by monitoring the curve of the loss function.
> We did not write $i$ as index. However, we now think that it is more readable to write $i$ as index and to replace $\theta_0$ with $\theta_i$ and $\theta$ with $\theta_{i+1}$, as you commented. We will revise the paper accordingly.
>
> **>>the pendulum task**
>
> This is a swing-up task with under-actuated pendulum. The goal is (angle,angular velocity) = $(0,0)$, which is an unstable equilibrium point if there is no control. The initial state is around $(\pi,0)$. Specifically, the initial state is $(\pi+\epsilon_1,\epsilon_2)$, where $\epsilon_1$ and $\epsilon_2$ are sampled from the Gaussian distribution with mean 0 and std 0.5. Figure 1 shows five trajectories. We will replace "stabilize at $(0,0)$" with "swing-up to $(0,0)$" and describe the above more clearly in the revised paper.
>
> **>>Minor typos**
>
> We will revise them accordingly. Thanks.

---

> > ### Comment · Reviewer_PdDh · 2021-08-18
> > **Evaluation task**
> >
> > Thank you for clarifying the control problem you are solving in the empirical evaluation section.

---

### Official Review · Reviewer_mD7J · 2021-07-10

**Rating:** 6
**Confidence:** 4

**Summary:**

The paper presents a model-based policy optimization algorithm for offline reinforcement learning. To account for the mismatch between the available dataset and the experience used during training, the proposed method uses estimation of the density ratio of the state distributions induced by the estimated and the real model. This importance weight, called “artificial”, is in contrast with the one, referred to as “natural”, employed by other off-policy methods involving the densities for two different policies under a same model. The resulting model learning objective is inserted into an expectation-maximization algorithm, which is then evaluated in a standard offline RL benchmark.

**Limitations And Societal Impact:**

The main limitations are mostly addressed in the related section of the paper. A question the authors should answer (and that could be a possible limitation for the method) is whether the advantages provided by the model learning procedure are only clear in the presence of model misspecification.

Given the mostly theoretical and algorithmic nature of the paper, I do not foresee any immediate societal impact.

**Main Review:**

The ideas presented in the paper are appealing: by leveraging a known bound on the performance difference under two different models, a method for model learning based on density ratio estimation techniques can be derived. Similarly, the same objective can be integrated into a policy optimization algorithm, which is reminiscent of recent model-based offline reinforcement learning approaches. The derivation of the algorithm is generally sound, although the practical approximation used in the experiments looses most of the guarantees of the initial counterpart.

**Originality**: The method is, to be the best of my knowledge, a new use of importance ratio estimation for model learning in offline RL. Nevertheless, some reference to relevant related work is missing.

**Quality**: The general quality of the work is mostly okay for the theoretical grounding, but less convincing for the experimental evaluation.

**Clarity**: The clarity of the writing could be improved.

**Significance:** The presented method is a new approach at the intersection between two very active research areas, decision-aware model-based RL and offline RL.

**Major concerns**:

- For the experiment on the D4RL domain, given only 4 runs and largely overlapping error bars, I do not think there is enough experimental evidence to argue that one method is better than the other on this particular domain. Even the explanation provided by the paper for the *Walker2d-medium dataset* is not very convincing for me: the learning process can cause the agent to fall for many other, confounding, reasons (e.g., the random initialization of the function approximators in use) apart from fundamental differences in the two methods and I do not see enough nuances in the experimental design to discern whether that one is the specific cause or not.
- Given the nice properties shown for the method in Section 6.1, I would have expected a clear difference in performance with respect to the baseline in Section 6.2. How can the lack of it be explained? Is the difference in performance in the first experiment due to the limited capacity of the model? What happens if more hidden units are used for the model of the dynamics?
- The exposition is often unclear for a generic reader. To just give two examples, in line 19, it is written that the paper focuses on offline MBRL, without being explicit on the meaning of this expression; in line 187, it is used the expression "policy evaluation", in a sense that is different from the one the typical RL researcher would expect (i.e., computing the value function for a policy).

**Minor concerns**:

- As reported in the related works, the method presented in the paper is related to previous work in decision-aware model learning. By contrast with what is written in the section, the use of a weighted log-likelihood objective for learning a model in the context of offline RL was already employed in (D'Oro, Pierluca, et al. "Gradient-aware model-based policy search." Proceedings of the AAAI Conference on Artificial Intelligence. 2020.), which incorporates a trajectory-based  importance weight to handle the mismatch between behavior and target policies.
- Can you elaborate again on the passage from Equation (3) to Equation (4)? This should not be so important for the paper overall, but is the importance weight (which is unbounded) missing from the second term? Also, in the same derivation, is an absolute value missing from the first inequality?

**Recommendation:** At the current state, I tend to recommend to reject the paper. I do not see enough experimental, nor theoretical, evidence about the advantages of the proposed method over existing baselines. I am open to increase my score if the authors adequately address my points of concern.

——

**After rebuttal:** I thank the authors for their response. After reading it, I tend to recommend to accept the paper. The bug disclosed and fixed by the authors does not change my general opinion, and my updated score.

**Time Spent Reviewing:**

6

---

> ### Author Response · Authors · 2021-08-10
> **Response to Reviewer mD7J**
>
> Thank you for your helpful comment.
>
> **>>Major concerns 1**
>
> After reading this comment, we check 10 runs for the Walker2d-medium and Walker2d-medium-expert datasets. The (normalized) scores are:
> * Walker2d-medium        ($\alpha=0.0$)   73.0 $\pm$ 17.2
> * Walker2d-medium        ($\alpha=0.2$)  63.3 $\pm$ 25.8
> * Walker2d-medium-expert ($\alpha=0.0$)  85.7 $\pm$ 38.4
> * Walker2d-medium-expert ($\alpha=0.2$)  108.2 $\pm$  2.4
>
> For Walker2d-medium-expert, $\alpha=0$ falls down for 2 runs out of 10 runs. The scores of the failed episodes are about 20 and 0, while the score of a successful episode is over 100. The large error bar comes from such a big difference in score between successful and failed episodes. As a result, the success rate is improved, while the error bars are overlapping. We think that the improvement of success rate is important.
>
> For Walker2d-medium, not only $\alpha=0.2$ but also $\alpha=0$ fall down for 2 runs out of 10 runs, due to the high variance of the real return. As a result, the difference in normalized scores is decreased. We still think that there is no special reason to choose $\alpha=0$.
>
>
>
> **>>Major concerns 2**
>
> To the question "Is the difference ...",  yes, this is due to the limited capacity of the model. For example, when learning 5-layer NNs with 128 hidden units based on unweighted ERM using the same hyperparameters, the learned model captures the swing-up behavior. For another larger model, we think that it can also predict the swing-up behavior by learning using appropriate hyperparameters.
>
> To the question "How can the lack of it be explained?", we cannot answer clearly yet. One of our guesses is that Section 6.1 requires interpolation, while Section 6.2 requires extrapolation. Importance-weighting is useful for near extrapolation, while far extrapolation requires another idea, for example, BNNs with a structured prior. Since further discussion is needed, we leave it a future work.
>
>
> **>>Major concerns 3**
>
> (line 19) We mean offline MBRL by MBRL that learns a policy from previously collected datasets. We will explicitly write this in the revised paper.
>
> (line 187) We mean "policy evaluation" by estimating the real expected return by the simulated expected return. In line 209, we compare the real and simulated returns, i.e. the result of policy evaluation. In this sense, we think that "policy evaluation" here is one of typical usages. However, thanks to your comment, we also notice that it may be difficult to understand that this experiment discusses policy evaluation, because the result of the policy evaluation itself is discussed at later part. Thus, we will revise the paper to discuss the result of policy evaluation first and the result of model estimation subsequently.
>
>
> **>>Minor concerns 1**
>
> Gradient-aware model-based policy search (GAMPS) uses the ratio of distributions of behavior and target policies on the real environment, which is called "the natural weight" in this paper. We differ from it in that we are discussing "the artificial weight", the distribution ratio of real and simulated data.
> Since GAMPS is an important existing work, we will refer it in the revised paper.
>
> **>>Minor concerns 2**
>
> The inequality for the second term is as follows. By definition, $-h(s,a)\le -h_{\min}$, and we have $E_{d^\pi_{P_{\theta}}}[-w^\pi_\theta(s,a)h(s,a)]\le E_{d^\pi_{P_\theta}}[-w^\pi_\theta(s,a)h_{\min}]$. Writing the expectation over $d^D_{P_\ast}$, we have $E_{d^\pi_{P_\theta}}[-w^\pi_\theta(s,a)h_{\min}] = E_{d^D_{P_\ast}}[-h_{\min}]$. Since $h_{\min}$ is not a random variable, we have $E_{d^D_{P_\ast}}[-h_{\min}]=-h_{\min}$. Finally, $E_{d^\pi_{P_{\theta}}}[-w^\pi_\theta(s,a)h(s,a)]\le -h_{\min}$.
>
> Thanks for the comment about absolute value. We will modify the expected value in the first inequality as $E[| \sum ... |]$.

---

### Official Review · Reviewer_L2Wz · 2021-07-16

**Rating:** 6
**Confidence:** 3

**Summary:**

This work studies how to estimate a learned model in offline model-based reinforcement learning (MBRL). The authors propose to use "the artificial weight," which is the ratio of offline data and simulated future data, to improve the model estimation. They further propose a new objective function for offline MBRL and an EM-style algorithm to optimize it. The authors evaluate their algorithm in several different tasks.

**Limitations And Societal Impact:**

Yes, the authors have adequately addressed the limitations and potential negative societal impact of their work.

**Main Review:**

The authors devise a new loss function for offline MBRL to alleviate the covariate shift issue. The core idea is weighting with the state-action distribution ratio of offline data and "simulated" future data, which is interesting.
Nevertheless, the authors may want to conduct several additional experiments to demonstrate the effectiveness of the proposed approach.
1. The authors may want to compare the proposed approach with that using "the natural weight." Several popular OPE methods can estimate the natural weight, such as DualDICE [1], GenDICE [2], and Gradient DICE [3].
2. The author may want to provide a comparison with state-of-the-art offline algorithms, such as CQL [4] and BEAR [5].
3. The results in Table 1 seem to show that using "the artificial weight" does not improve the performance. I wonder whether the proposed method reduces the error of the fictitious samples. The authors may want to provide more details to explain the results in Table 1. Is it possible the weight $[w^\pi_{\theta_0}(s_n,a_n)]^\alpha$ is so close to 1 that its effect on the learned model is negligible?
3. The authors may want to show how the hyperparameter $\alpha$ affects the accuracy of the learned model. For example, the authors may want to illustrate the K-step prediction error during the generation of fictitious samples when $\alpha$ varies.

[1] Nachum, O., Chow, Y., Dai, B., and Li, L. DualDICE: Behavior-agnostic estimation of discounted stationary distribution corrections. arXiv:1906.04733, 2019.

[2] Zhang, R., Dai, B., Li, L., and Schuurmans, D. GenDICE: Generalized ofﬂine estimation of stationary values. arXiv:2002.09072, 2020.

[3] Zhang, S., Liu, B., and Whiteson, S. Gradientdice: Rethinking generalized ofﬂine estimation of stationary values. In International Conference on Machine Learning, 2020.

[4] Kumar A, Zhou A, Tucker G, et al. Conservative q-learning for offline reinforcement learning. arXiv:2006.04779, 2020.

[5] Kumar A, Fu J, Tucker G, et al. Stabilizing off-policy q-learning via bootstrapping error reduction. arXiv:1906.00949, 2019.

**Time Spent Reviewing:**

10 hours.

---

> ### Author Response · Authors · 2021-08-10
> **Response to Reviewer L2Wz**
>
> Thank you for your helpful comment.
>
> **>>1**
>
> The results of DualDICE, GenDICE, and Gradient DICE on the D4RL MuJoCo dataset do not seem reported currently. We need to conduct the experiments or to apply our algorithm to their setting. However, we could not do that due to the time limit. We are sorry for this.
>
> **>>2**
>
> The results of CQL and BEAR on the D4RL MuJoCo dataset is shown in Table 1 of Kumar et al. 2020. Roughly, our method is worse for the walker2d-medium dataset, similar for the medium-expert datasets, and better for the other datasets, compared to CQL. At the same time, our MOPO instantiation (i.e. $\alpha=0$) is worse for the walker2d-medium-expert dataset, similar for the medium-expert datasets, and better for the other datasets.
>
> **>>3,4**
>
> Section 12.3 in the supplementary material discusses the result for the Walker2d-medium-expert dataset when varying $\alpha$. Using $\alpha=1.0$ does not learn a good policy. Using $\alpha=0.5$ is better than using $\alpha=1.0$, but the gap between simulated and real return seems larger than using $\alpha=0.2$. Using $\alpha=0.1$, in which the weight is closer to 1, seems to have the same problem with using $\alpha=0$. These results show how $\alpha$ affects the learned policy. Table in Section 12.3 shows the weighted log likelihood, which measures the accuracy of the learned model. Here, $\alpha=0.2$ is the best.
> The result when varying $\alpha$ shows an example where $\alpha$ controls the trade-off between consistency and stability described in line 136.

---

> > ### Comment · Reviewer_L2Wz · 2021-08-19
> > **Thanks for the response**
> >
> > Thanks for the response. The current response does not properly address my main concerns. The authors' claim---"the nature weight is not easy to use"---is not convincing for me, as several OPE techniques (such as GenDICE, Gradient DICE) have been successfully used to estimate "the nature weight" in Open Gym tasks. Though the comparison in D4RL tasks may be difficult due to the time limitation, the authors may still need to make an effort to provide a comparison in a toy environment, such as that used in Section 4.1. If the author can empirically show the advantages of "the artificial weight" over "the natural weight," I will be happy to raise my score.

---

> > > ### Author Response · Authors · 2021-08-25
> > > **Additional experiment**
> > >
> > > We would like to compare our method with GradientDICE on the off-policy evaluation (OPE) experiment in Reacher-v2 in [1]. Due to the time limit, we discuss only the case of $\gamma=0.9$. For GradientDICE, we use the authors' code and hyperparameters and get the result after $10^4$ steps. For empirical risk minimization (ERM) and Algorithm 2 in our paper, we use the same model and hyperparameters with Section 6.2 in our paper. To compute the expected return, we run the author's code five times and average it. The computed expected return is $-0.250$.
> > >
> > >
> > > Firstly, we show the results when using the same offline data with [1], where the results are averaged over 5 runs. Here, all methods seem to estimate the expected return with similar accuracy.
> > >
> > >
> > > |  method  |  estimated value  |
> > > | ---- | ---- |
> > > |  GradientDICE  |  $-0.224$  |
> > > |  ERM  |  $-0.265$  |
> > > |  Algorithm 2 ($\alpha=0.2$)  |  $-0.263$  |
> > >
> > >
> > >
> > > Secondly, we show the results when using another offline data collected using a random policy. Here, GradientDICE seems to estimate the expected return inaccurately.
> > >
> > > |  method  |  estimated value  |
> > > | ---- | ---- |
> > > |  GradientDICE  | $-0.050$  |
> > > |  ERM  |  $-0.252$  |
> > > |  Algorithm 2 ($\alpha=0.2$)  |  $-0.255$  |
> > >
> > >
> > > GradientDICE is an importance sampling (IS)-based method for OPE. One of challenges for IS-based methods is that: when the behavior policy is too different from the target policy, the importance weights will become degenerate, and any estimate of the return or the gradient will have too much variance, see Section 3.5 in [2]. The difference between behavior and target policies is small in the first experiment but is large in the second experiment. Thus, the result of GradientDICE in the second experiment is due to this issue.
> > >
> > > ERM is the direct method (DM) for OPE. The advantage is its relatively low variance, while the disadvantage is the bias due to the modeling error.
> > >
> > > Algorithm 2 is a variant of the DM for OPE. Although the artificial weight may also have high variance as described above, Algorithm 2 can soften the effect by choosing $\alpha$. That is, Algorithm 2 can also enjoy the advantage of the DM by choosing $\alpha$. Note that softening the artificial weight is justified as stabilizing model fitting, see line136 in page4. Thus, the results of ERM and Algorithm 2 in the second experiment are due to the advantage.
> > >
> > > [1] Zhang et al. GradientDICE: Rethinking Generalized Offline Estimation of Stationary Values, 2020.
> > >
> > > [2] Levine et al. Offline Reinforcement Learning: Tutorial, Review, and Perspectives on Open Problems, 2020.

---

> > > > ### Comment · Reviewer_L2Wz · 2021-08-26
> > > > **I will raise my score from 5 to 6,**
> > > >
> > > > Thanks for the response. The authors' response has largely addressed my major concern about the comparison with advanced OPE methods. Thus, I will raise my score from 5 to 6, more in favor of accepting the paper.

---

### Official Review · Reviewer_Ty7z · 2021-07-21

**Rating:** 7
**Confidence:** 3

**Summary:**

This paper presents a model-based algorithm for offline reinforcement learning (RL) that attempts to address the covariate shift issue by re-weighting the model losses for different datapoints. Based on an upper bound of the policy evaluation error, a joint loss function is defined over both the policy and the model parameters, and the two are optimized in an alternating fashion. Several simplifications and tricks are described to make the algorithm computationally tractable.

On a simple pendulum environment, the proposed algorithm predicts the policy value much more accurately than a model based on empirical risk minimization (ERM). It also obtained good results on the MuJoCo tasks from the D4RL benchmark.

**Limitations And Societal Impact:**

Yes

**Main Review:**

The algorithm in the paper is novel to my knowledge. Despite some similarities to MOPO, it approaches the model estimation differently (using weighted losses and training jointly with the policy), leading to a more complicated algorithm but better results on D4RL tasks. Considering that distribution shift is a major challenge in offline RL, this strikes me as a useful contribution.

Given that the algorithm requires a number of simplifications and tricks, it would be useful to ablate those design choices where possible. For example, the authors take a majorization-minimization approach “To make each optimization step easier”, but give no explanation as to what goes wrong if no MM surrogate is used. Does it diverge? Or are the results just not as good?

The algorithm’s claimed performance on the D4RL MuJoCo tasks is strong compared to previous algorithms. However, I had to consult the D4RL paper to determine this. I would recommend putting one or more of the baseline algorithms in the table for ease of comparison.

I found it odd that numerical results on D4RL were not presented for the simplified version of the algorithm. It appears to work similarly to the full version on the pendulum environment as depicted in Figure 1(b), but it would be good to verify that it scales to more challenging environments, especially given that its primary motivation (relative to the full version) is to reduce the computational burden.

The writing of the paper is understandable, although the grammar could be improved in many places, and there are several typos.

**Time Spent Reviewing:**

4

---

> ### Author Response · Authors · 2021-08-10
> **Response to Reviewer Ty7z**
>
> Thank you for your helpful comment.
>
>
> **>>majorization-minimization**
>
> Firstly, using MM, updating $\theta$ reduces to optimizing Equation (16), which is the same form as weighted ERM in Section 5.1. We can apply the algorithm proposed in Section 5.1 with a small modification. If not using MM, we need a more complicated algorithm. Secondly, using MM, updating $\pi$ results in optimizing Equation (17), which is planning in a MDP. To optimize it, we can take advantage of an existing efficient dynamic programming or reinforcement learning algorithm. If not using MM, the optimization objective is not the expected return, and we cannot use such an efficient algorithm. Although to directly optimize Equation (8) is ideal, we have no efficient algorithm for it currently.
>
> If you found another under-explained simplification, we would like to know that.
>
>
> **>>putting one or more of the baseline algorithms**
>
> We will add the reported scores of other baseline algorithms, e.g. CQL and the original MOPO implementation.
>
>
> **>>numerical results on D4RL**
>
> We use only the simplified version in numerical experiments on D4RL, as applying the full version to such large-scale tasks is computationally unrealistic. We will revise the description to state it clearly.

---

> > ### Comment · Reviewer_Ty7z · 2021-09-01
> > **Response to authors**
> >
> > Thank you for clarifying my confusions!

---

### Author Response · Authors · 2021-08-25
**fix bug with minor effects**

We found a bug in our code while trying the additional experiments required. However, we think that it has minor effects on our results for the following reason.

The bug was how to clip reward in Section 6.2. We wanted to clip the predicted reward to $[-10 |R|, 10|R|]$, where $|R|$ is the maximum absolute value of the reward data (page8, line235).　Actually, due to the opposite inequality sign at line46 in `model_env/e_step.py`, we clipped the predicted reward to $[-c |R|, c|R|]$, where $c$ depended on dataset. There was no problem for $c>1$. However, for $c<1$, there was a problem where not only some predicted reword but also some reward data were clipped.

The problematic cases were hopper-medium and hopper-medium-expert. Specifically, $a=0.92$ for hopper-medium, and $a=0.83$ for hopper-medium-expert. Note that these cases were not important for our discussion, as the scores of $\alpha=0$ and $\alpha=0.2$ were similar.

After fixing the bug, we re-run the experiments for Table 2 (with 5 runs). The results are below.

|  dataset  |  $\alpha=0$  | $\alpha=0.2$  |
| ---- | ---- | ---- |
| halfcheetah-random  | 48.7 $\pm$ 2.8  | 49.1 $\pm$ 3.2  |
| halfcheetah-medium  | 75.7$\pm$ 1.5   | 73.1$\pm$ 5.2   |
| halfcheetah-medium-replay  | 72.1 $\pm$ 1.4 |  65.5 $\pm$ 6.4  |
| halfcheetah-medium-expert  | 73.9 $\pm$ 24.2  | 85.7 $\pm$ 21.6  |
| hopper-random  | 30.2 $\pm$ 4.4  |  32.7 $\pm$ 0.5  |
| hopper-medium  | 100.9 $\pm$ 2.7  | 104.1 $\pm$ 1.2  |
| hopper-medium-replay  | 97.2 $\pm$ 10.9  | 104.0 $\pm$ 3.2  |
| hopper-medium-expert  | 109.3 $\pm$ 1.1  | 104.9 $\pm$ 10.1 |
| walker2d-random  | 16.5 $\pm$ 6.6 | 18.4 $\pm$ 7.6  |
| walker2d-medium  | 81.7 $\pm$ 1.2  | 60.7 $\pm$ 29.0 |
| walker2d-medium-replay  | 80.7 $\pm$ 3.1  |  82.7 $\pm$ 3.3 |
| walker2d-medium-expert  | 59.5 $\pm$ 49.4  | 108.2 $\pm$ 0.54 |

In the above table, the case of $\alpha=0.2$ is better for walker2d-medium-expert and worse for walker2d-medium. For the other cases, $\alpha=0.2$ and $\alpha=0.0$ have similar scores. In these respects, the above table and Table 2 are similar. Similar figures to Figure 2 are also obtained.

Thus, we think that the past discussions are still valid and that the bug has minor effects.

---

> ### Comment · Area_Chair_xH5B · 2021-08-30
> **Thank you. We will consider this.**
>
> Dear authors,
>
> Thank you for reporting the bug. I have already asked the reviewers to re-evaluate your paper in light of this new data.
>
> - In your comments above, does a = 0.92 or 0.83 refer to the dataset dependent clipping threshold c (the paragraph before)?
> - If possible, would you please create an anonymous link with relevant figure(s) (Figure 2), in case reviewers want to take a look at the updated figure?
>
> Thank you,
> Area Chair

---

> > ### Author Response · Authors · 2021-08-31
> > **Thank you for the reconsideration**
> >
> > Thank you for the reconsideration.
> >
> > >In your comments above
> >
> > Yes. They should be $c=0.92$ and $c=0.83$. Thank you for the correction.
> >
> > >If possible
> >
> > Thank you for the comment. Please check https://file.io/myk1PjQBzycZ

---

### Public Comment · ~Shentao_Yang1 · 2022-01-07
**Some Curiosities about the Experimental Section**

Dear authors,

It is my honor to read your illuminating paper! Regarding the Experimental section, I have the following curiosities:

1. You mentioned that if $\alpha = 0$, the proposed algorithms can be seen as an instantiation of MOPO. I do observe, however, that the results for the $\alpha = 0$ version is significantly better than those of the ``original MOPO", which you cited from Table 1 of the MOPO paper. May I ask whether you have any thoughts for such a performance discrepancy between the two MOPO instantiations (the original one and yours)? In particular, which components in your $\alpha = 0$ version improves upon the original MOPO instantiation?

2. I notice the following algorithmic detail in your description of the algorithm used for the D4RL experiments:
> To stabilize the M-step, the agent terminates a simulation episode when a state or action variable diverges.

To the best of my knowledge, this is a technical innovation compared to the original MOPO. Could you provide more details on how you implement this technical idea?

3. In the appendix, you wrote:
> The number of times to perform SAC update per M-step is $5 × 10^5$.

I am sorry that I am a bit confused about this sentence. Does it means that each M-step contains $5 × 10^5$ mini-batch stochastic gradient descent updates to the policy $\pi$?

4. I notice that for the D4RL experiments you iterates the main loop of Algorithm 3 twice. To my understanding, this may not show the full potential of the proposed algorithm since the first iterate of the main loop will estimate the transition $\mathcal{P}_\theta$ using ERM, which is what MOPO-style MBRL algorithms do. The estimated $\mathcal{P}_\theta$ is subsequently refined only once, in the next/final iterate of the main loop.
In viewing this, have you tried any experiments that increase the iterations of the main loop, possible accompanied by a corresponding decrease in the number of policy updates per M-step to keep the total number of policy updates unchanged? Otherwise, do you have any intuition why iterating the main loop of Algorithm 3 twice is sufficient?

Looking forward to your responses!

Thanks,

Shentao

---

> ### Public Comment · Authors · 2022-01-10
> **Thanks for the comment**
>
> Thank you very much for your interest in our paper.
>
> >1
>
> The difference we are aware of is how to generate simulation data used in the SAC update. The original MOPO implementation generates it by $h$-step rollout simulation starting from states drawn from offline data, where $h$ is a hyperparameter, see the Appendix of the MOPO paper.  Our implementation generates it by rollout simulation starting from states drawn from the initial state disribution, where the rollout length is randomly chosen according to discount factor $\gamma$, see the definition of $\hat{D}^\pi_\theta$ in Section 3. We think that this is the largest difference that leads to the difference of the scores, though other components might also have effects.
>
> >2
>
> We think that this idea is still just a heuristics, as we have not discussed enough in the context of MBRL research. The motive for using it is as follows. Due to the above-mentioned implementation difference, the simualation rollout length in our implementation becomes longer, compared to the original implementation. The magnitude of simulated reward often becomes unreasonablly large, compared to offline data. The SAC update using dataset including such rewards may be unstable, especially in the early stages. To mitigate it, we stop rollout simulation if the magnitude of reward exceeds a heuristically determined threshold. We also think that the original MOPO implementation will not suffer from such unreasonable rewards, as it uses shorter rollout lengths.
>
> This discussion seems related to the question of ``when to trust models.'' We believe that it is still one of the issue to challenge.
>
> >3
>
> Yes, it does. We are sorry for our confusing sentence.
>
> >4
>
> We have tried iterating three times in preliminary experiments for some of D4RL MuJoCo benchmark datasets. Although we have not made enough comparisons, it seemed that the scores and discussion would not change so much.
>
> Since our method is an EM-style optimization using one objective function, the number of main-loop iterations will be determined by monitoring the objective function, if computational resources are enough. However, it is difficult to determine the number of main-loop iterations according to the available computational resources. For example, for medium-expert dataset in this experiment, the E- and M- steps require similar calculation times. If increasing the iterations of main-loop while using the same computational resources, then the calculation time for E-step increases, and we should reduce the total number of SAC update accordingly. How to choose the number of main-loop iterations in general is an important question, but we cannot answer it for now.

---

### Decision · Program_Chairs · 2021-09-27

**Decision:**

Accept (Poster)

**Comment:**

The reviewers find the problem addressed by the paper important and the proposed idea novel. The most important negative aspect of the paper, however, is with its empirical studies:

There are several issues there:

1) Although the empirical results for the simpler environment (Pendulum) is promising, the results for more difficult set of problems (MuJuCo tasks) are underwhelming. In most cases, there is no significant difference between the proposed method (which corresponds to the choice of $\alpha > 0$, and specifically $\alpha = 0.2$ in the results), and $\alpha = 0$, which would be the case when the method reduces to the usual ERM with no sample weighting.

The lack of significant difference is more exasperated as the results are based on only 5 random seeds and the standard deviation is  large in most cases, and we have a lot of overlap between confidence intervals. It is difficult to say with certainty if the new method actually helps or not. It also appears that there is no really good explanation on why there is not much difference.

I should note that the authors provided new results during the discussion period with 10 runs for a subset of the problems.

Although it is OK if the method is not performing very well, it would make a much stronger paper if we at least learn why that is the case.
I suggest that the authors perform similar experiments, but with larger number of runs and perhaps with proper statistical tests.


In addition to this major issue, there are some other comments:

2) The proposed method follows a sequence of approximation. Ablation studies would be helpful in providing insight in the effect of each approximation.

3) Empirical comparison with methods that use "natural" weights strengthens the paper.

4) There is another comment, which was not raised by reviewers, but is based on my reading of the paper. I do not give much weight to this because the authors have not had a chance to respond to it.

In the derivations leading to Eq. (3), the first inequality uses the telescoping lemma with the choice of distribution of expectation to be w.r.t. the discounted future state distribution induced by the model $P_\theta$ and the value function inside the expectation to be w.r.t. the true model $P^*$. This derivation leads to the introduction of density ratio w_theta in order to change the outer expectation from $d_{P_\theta}$ to $d_{P^*}$ in Eq. (3).

But I believe it was also possible to use the telescoping lemma differently.
If the expectation was taken w.r.t. $d_{P^*}$, the value function inside the expectation would be $V_{P_\theta}$. In the next line, we would still upper bound that value function by $\max_{s,a} |r(s,a)| / (1 - \gamma)$, and we would get the same $B$. The result would be that we wouldn't need any density ratio $w_\theta(s, a)$.

If what I wrote is correct, this means that we could have an upper bound that does not require the introduction of any weights. If so, much of the proposed method would not be needed.

It would be helpful if the authors can clarify this. Perhaps one upper bound is tighter than the other one?

**Evaluation:**
Overall, I would consider this as a *borderline* paper. The suggested method is interesting and might be useful too, but I believe that there is room for improvement and there are straightforward ways that this paper can significantly be improved.